# UltraMedical: Building Specialized Generalists in Biomedicine

**Kaiyan Zhang**[α,ε]    **Sihang Zeng**[β]    **Ermo Hua**[α,ε]    **Ning Ding**[α*]    **Zhang-Ren Chen**[γ]
**Zhiyuan Ma**[α]    **Haoxin Li**[α]    **Ganqu Cui**[α]    **Biqing Qi**[α]    **Xuekai Zhu**[δ]    **Xingtai Lv**[α,ε]
**Jin-Fang Hu**[γ]    **Zhiyuan Liu**[α]    **Bowen Zhou**[α*]

[α] Tsinghua University    [β] University of Washington
[γ] The First Affiliated Hospital of Nanchang University
[δ] Shanghai Jiao Tong University    [ε] Frontis.AI

*zhang-ky22@mails.tsinghua.edu.cn*
*{dn97, zhoubowen}@tsinghua.edu.cn*

## Abstract

Large Language Models (LLMs) have demonstrated remarkable capabilities across various domains and are moving towards more specialized areas. Recent advanced proprietary models such as GPT-4 and Gemini have achieved significant advancements in biomedicine, which have also raised privacy and security challenges. The construction of specialized generalists hinges largely on high-quality datasets, enhanced by techniques like supervised fine-tuning and reinforcement learning from human or AI feedback, and direct preference optimization. However, these leading technologies (e.g., preference learning) are still significantly limited in the open source community due to the scarcity of specialized data. In this paper, we present the UltraMedical collections, which consist of high-quality manual and synthetic datasets in the biomedicine domain, featuring preference annotations across multiple advanced LLMs. By utilizing these datasets, we fine-tune a suite of specialized medical models based on Llama-3 series, demonstrating breathtaking capabilities across various medical benchmarks. Moreover, we develop powerful reward models skilled in biomedical and general reward benchmark, enhancing further online preference learning within the biomedical LLM community.
GitHub: https://github.com/TsinghuaC3I/UltraMedical
Huggingface: https://hf.co/collections/TsinghuaC3I

## 1    Introduction

The advent of Large Language Models (LLMs) has brought forth numerous potential applications in the field of biomedicine and healthcare, encompassing medical education, clinical practice, and scientific research. Recent studies suggest that proprietary models such as GPT-4, Med PaLM 2, and MedGemini have the potential to function as integrated medical generalists [46, 59, 76], even achieving expert-level performance on some medical benchmarks. In the meantime, although there have been advancements, open-source LLMs fine-tuned on synthetic medical instructions still significantly lag behind proprietary models [71, 20, 10, 51, 31].

Despite the remarkable capabilities, proprietary models may face security and privacy challenges due to the sensitive nature of medical data, such as potential data breaches and the risk of exposing sensitive patient information [35, 80, 40]. On the other hand, open-source LLMs can be customized

---

*Corresponding Author.

38th Conference on Neural Information Processing Systems (NeurIPS 2024) Track on Datasets and Benchmarks.

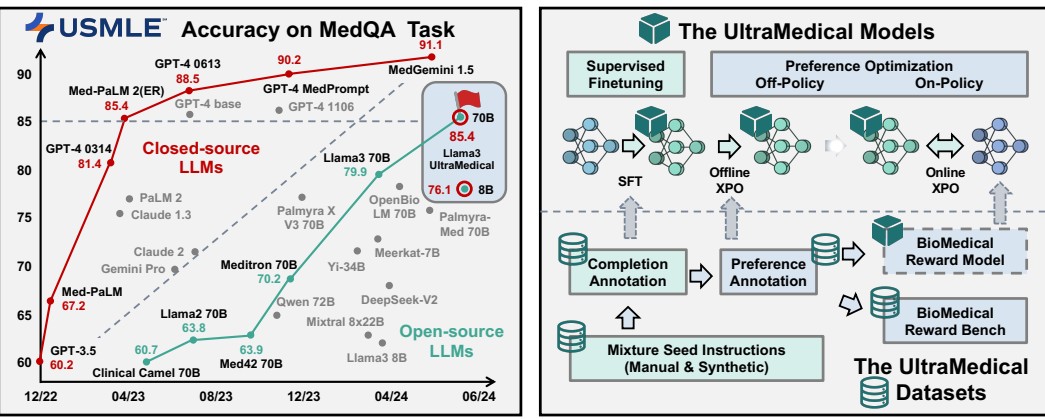

Figure 1: The UltraMedical Datasets, Models and Performance on MedQA.

and adapted to specific healthcare contexts by fine-tuning on local datasets, enabling the development of models tailored to the needs of specific patient populations, healthcare settings, or research questions, thereby enhancing their practical utility and impact. Exploring how to build open-source, GPT-4-level LLMs in the field of biomedicine is underway. Beyond supervised fine-tuning, preference learning technologies like Reinforcement Learning from Human or AI Feedback(RLHF or RLAIF) [33, 29], direct preference optimization (DPO) [52], Kahneman-Tversky Optimization (KTO) [17] and others [73, 23, 24, 43] has proven to play a significant role in enhancing the reasoning abilities of open LLMs in various tasks such as coding, mathematics, and logic [44, 78]. However, preference learning remains under-explored in the biomedical community [77], which is mainly limited by the scarcity of high-quality and extensively annotated preference datasets.

In this paper, we investigate the development of specialized generalists in the field of biomedicine from a data-centric perspective. We first construct a large-scale, diverse, and high-quality dataset by combining manual and synthetic biomedical instructions, which comprise medical exam problems, PubMed literature research, and open-ended questions. We then build on the outputs of various LLMs to painstakingly annotate these instructions, along with corresponding preference scores and rankings, to ultimately create our UltraMedical dataset. By leveraging UltraMedical and previous open-domain datasets such as UltraChat [15], we further explore how to fuse professional skills with general skills and then fine-tune the Llama-3 family of models to produce competitive medical models. Additionally, we train a reward model based on UltraMedical preferences annotations and previous feedback datasets [13, 19, 78] achieving advanced results in both our annotated medical benchmark and RewardBench [32]. Based on the preferences of the constructed reward models, we continuously optimize the UltraMedical LMs through a self-generated response strategy, and finally result in more powerful models. Finally, our 8B model significantly outperforms previous larger models such as MedPaLM 1 [58], Gemini-1.0 [64], GPT-3.5, and Meditron-70B [9] in terms of average score on popular medical benchmarks. Moreover, our 70B model achieved an 86.5 on MedQA-USMLE, marking the highest result among open-source LLMs and comparable to MedPaLM 2 [59] and GPT-4.

Specifically, our paper makes the following contributions:

- We construct the UltraMedical collections, a high-quality collection of about 410K medical instructions that adhere to principles of complexity and diversity. This dataset combines manual and synthetic prompts. A subset of approximately 100K instructions within UltraMedical has been annotated with preferences over completions from advanced medical and general models, contributing to fine-tuning, reward modeling, and preference learning.

- By fine-tuning the Llama-3 series on UltraMedical using a multi-step optimization strategy, as described in § 3, we achieved competitive results in open-source medical benchmarks with Llama-3-8B/70B, detailed in § 4. The results indicate that we can narrow the gap between open-source and proprietary models using the UltraMedical collections.

- Buiding upon UltraMedical preference data, we annotate the medical reward bench with the help of biomedical experts in § 3. We also pioneer the training of reward models in biomedicine based on UltraMedical preferences, resulting in advanced performance on

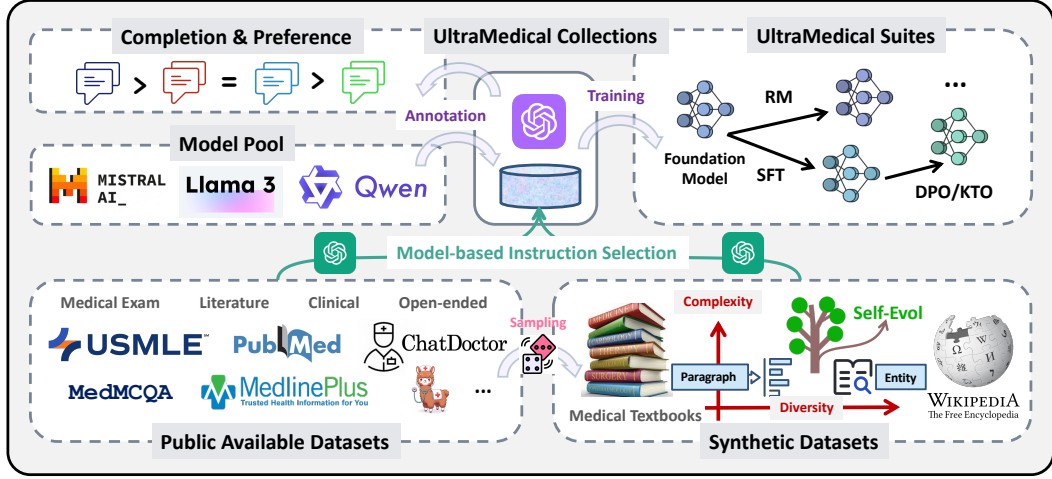

Figure 2: The Construction Pipeline for the UltraMedical Dataset.

both annotated medical and general reward benchmarks in § 5. This initiative significantly contributes to further online or iterative preference learning in this field.

- We release our datasets and our models to the public on both GitHub and Huggingface, aiming to foster collaboration and accelerate progress in the field of biomedical generative AI by providing valuable resources to the research community.

## 2 The UltraMedical Dataset

The UltraMedical dataset initially comprises a large-scale collection of approximately 410,000 high-quality medical instructions that combine manual and synthetic prompts. These prompts are partially created by us and selected from open sources, which are produced from the guidance of principles of diversity and quality. Secondly, the dataset includes about 110,000 instructions annotated with completions from various LLMs with preferences annotated by GPT-4. Thirdly, a subset of approximately 900 model-annotated preference pairs has been reviewed and corrected by human experts, forming the basis of the medical reward benchmark. In the following sections, we will first introduce the details of the UltraMedical collections as shown in Figure 2, including instruction composition in § 2.1 and data annotations in § 2.2, and dataset statistics in § 2.3, respectively.

### 2.1 Instruction Composition

#### 2.1.1 Principle of Diversity

UltraMedical comprises a variety of question types, including medical exam questions, literature-based questions, and open-ended instructions (clinical questions, research questions, and others). It comprises 10 manual and synthetic datasets. For publicly available datasets, we have gathered questions from multiple sources, including medical exams, medical literature, clinical questions, and open-ended instructions. These datasets feature not only manually curated instructions but also prompted instructions from GPT-4. The various data sources preliminarily enable the diversity principle of the UltraMedical dataset.

In addition to public datasets, we have created three synthetic datasets to augment the UltraMedical collection. Due to the high quality of questions in MedQA [27], we regard MedQA questions as a primary seed source. The first dataset, MedQA-Evol, is synthesized and evolved from the original MedQA data. The second dataset, TextBookQA, consists of multiple-choice questions derived from medical textbooks, using questions from MedQA as in-context examples. The last dataset, WikiInstruct, aggregates thousands of biomedical concepts from Wikipedia pages and expands them into more detailed knowledge and instructions. As visualized on Nomic AI Atlas in Figure 3, the diversity of the topics in the UltraMedical prompts validates the effectiveness of the aforementioned process. We provide details about each data source along with examples in the Appendix C and E.

Table 1: Instructions Statistics. Datasets marked with "★" represent our customized synthetic data, while the others are adapted from publicly available data. Average length and score by ChatGPT noted as *Avg.Len* and *Avg.Score*.

| Category | Synthetic | Dataset | # Original | Avg.Len | Avg.Score | # Retained |
|---|---|---|---|---|---|---|
| Examination | ✗ | MedQA | 10.2K | 128.94 | 7.35 | 9.3K |
| | ✗ | MedMCQA | 183K | 23.12 | 4.73 | 59K |
| | ✓ | ★ MedQA-Evol | 51.8K | 76.52 | 8.07 | 51.8K |
| | ✓ | ★ TextBookQA | 91.7K | 75.92 | 7.72 | 91.7K |
| Literature | ✗ | PubMedQA | 211K | 218.2 | 7.95 | 88.7K |
| Open-ended | ✗ | ChatDoctor | 100K | 98.93 | 6.83 | 31.1K |
| | ✗ | MedQuad | 47K | 8.21 | 4.54 | 6K |
| | ✓ | MedInstruct-52K | 52K | 36.05 | 5.25 | 23K |
| | ✓ | MedIns-120K | 120K | 84.93 | 5.36 | 25K |
| | ✓ | ★ WikiInstruct | 23K | 46.73 | 8.8 | 23K |
| ☆ **UltraMedical (Mixed)** | | **Instructions** | - | 101.63 | 8.2 | **410K** |
| | | **Preference Pairs** | 1.8M | - | - | **100K** |

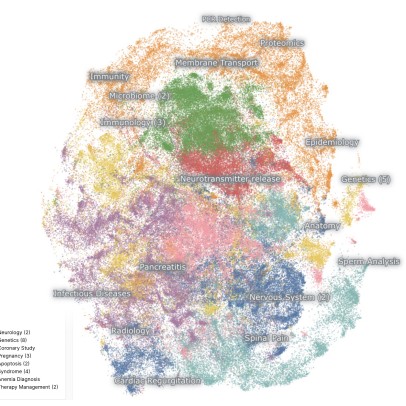

Figure 3: Broad Topics Distribution

### 2.1.2 Principle of Complexity

Beyond the diversity characteristic, UltraMedical also upholds the principle of complexity to inject knowledge and enhance reasoning abilities through complex instructions. There are primarily two routes to enhance the complexity of instructions, either pre-hoc or post-hoc. The former involves starting with various seed instructions to synthesize new instructions, followed by employing self-evolution on these synthetic instructions [72, 41]. The latter involves filtering instructions using heuristic rules or model-based rankers to select the most complex instructions [8, 81].

During the construction of the UltraMedical dataset, we employ both pre-hoc and post-hoc methods to enhance the complexity of the instructions. For publicly available datasets, we use `gpt-3.5-turbo` to assign a scale score ranging from 1 to 10 to each instruction, where 1 indicates an instruction that is easy to answer and 10 denotes one that is challenging for ChatGPT. For our synthetic dataset, we combine pre-hoc and post-hoc methods to ensure the complexity of the instructions. Initially, we implement a two-step self-evolution process on all synthetic instructions, and then further filter them based on model-derived scores. As illustrated in Table 1, there exists a strong correlation between the length and scores of instructions, with longer instructions often containing more entities and requiring the assistant to reason over context. However, direct linear relationship is not observed between these two metrics. Despite this, it is still necessary to employ a judger to filter out poor-quality instructions, even if they are lengthy. This finding is consistent with previous works [60, 82].

## 2.2 Data Annotation

### 2.2.1 Completions Annotation

After compiling diverse instructions, we annotate answers using `gpt-4-turbo` to optimize these responses for SFT. For multiple-choice questions, the chain-of-thought (CoT) [69] method has proven effective in distilling knowledge from large to small language models. Therefore, we instruct `gpt-4-turbo` to sequentially answer each question. Subsequently, we verify the answers against the ground truth and filter out incorrect responses. For incorrect answers, we further engage `gpt-4-turbo` with dynamically retrieved few-shot CoT examples from our annotated database. This process enables us to maximize the number of potential candidate samples while ensuring the quality of the completions.

### 2.2.2 Preference Annotation

Recently, an increasing number of studies have committed to building preferences in both general and specialized domains such as mathematics and coding. The primary strategy for obtaining completion candidates include: sampling several models from a mixed-scale model pool to compose completion candidates, sampling responses from a powerful base model and GPT-4, or simply sampling from the SFT model. There is no conclusive evidence to determine which strategy is the most effective. We sample responses from the top-tier open-source and proprietary models for preference annotation. For proprietary models, we just adapt `gpt-3.5-turbo` and `gpt-4-turbo`. For open-source

models, we select `Llama-3-8B/70B` [2], `Qwen1.5-72B` [5], `Mixtral-8x7B/22B` [26], along with our supervised finetuned UltraMedical 8B model. Subsequently, we use GPT-4 to rank the candidates based on score and explanation. However, there may be a bias in GPT-4 towards its own responses [50, 75]. Therefore, we choose the newest version of GPT-4 to score the completions, which is `gpt-4-2024-04-09`. More scalable and reliable annotation methods, such as fact-checking with search tools [70], could be employed, and we leave this exploration for future work.

**Preference Binarization:** For subsequent preference learning like DPO, binarization of preferences is necessary, involving a pair comprising a "chosen" and a "rejected" completion for each sample. Following the Zephyr protocol [66], the highest-ranked completion is selected as the "chosen" one. In instances where multiple completions share the top ranking or scores, the completion from GPT-4 is favored. Subsequently, a random completion from the remaining entries, excluding the top-ranked ones, is designated as the "rejected" completion.

**Medical RewardBench:** Drawing inspiration from RewardBench [32], which evaluates reward models using a variety of prompts and paired responses, we build *Medical RewardBench*. First, we randomly select 1,000 samples from all preference samples and set them aside from the training data. We then categorized the 1,000 samples into "easy", "hard", and "length" pairs according to the model's scores from GPT-4, while 100 samples for each sub-task. Finally, we obtain pairs for annotation and corrected the preferences with scores and ranks from GPT-4. To ensure the accuracy of the preference pairs, we engage biomedical clinicians, graduate students, and researchers in correcting the preferences. Beyond the Easy, Hard, and Length sets, we also allocate a portion of the samples to the Human set, which consists of samples revised by humans and potentially presents greater challenges. Further discussion is presented in § 5 and Appendix C.3.

**Human Annotation:** To ensure the reliability of the medical reward benchmark, we assembled a team of three experts, each with at least three years of research experience in biomedicine. They utilized a customized WebUI and academic search engines to validate question-answer pairs. For the reward benchmark, out of 1,000 test samples, only about 780 were retained where at least two annotators agreed on the same label. Samples with disagreements or both incorrect answers were removed. We provide more details about human annotation in Appendix C.4.

**Annotation Cost:** The costs associated with creating the dataset and benchmark primarily include GPT-4-Turbo API (version 1106) calls for instruction synthesis and response generation, as well as preference annotation, totaling approximately $20,000.

## 2.3 Dataset Statistics

**Overall:** As illustrated in Table 1, the UltraMedical collections ultimately comprise 410K instructions. For the preference annotation, we select the instructions with the highest scores from each dataset, resulting in approximately 100K instructions accompanied by eight models' completions. During the preference binarization process, we aim to maximize the selection, achieving $C_8^2 = 28$ combinations of "chosen" and "rejected" completions per instruction. Although we retain only completions with differing scores, we ultimately obtain approximately 1.8M pairs for reward modeling (approximately 18 times the size of the instruction.). We provide more details in Appendix C.

**Medical RewardBench:** For the initially given 1,000 test pairs, we ultimately retained 777 pairs following human expert annotation. These include 238 easy, 196 hard, 180 length-based, and 163 human-judged pairs. Approximately 233 pairs were filtered out due to issues such as incorrect formulations, difficulty in answering, or both. The human category comprises pairs where preferences differ between human annotators and GPT-4, which is regraded as even hard for GPT-4 to recognize.

## 3 The UltraMedical Suites

Based on the UltraMedical datasets, we develop the UltraMedical LMs and a reward model (RM) based on Llama-3 models using the following four steps: supervised fine-tuning in § 3.1, preference learning in § 3.2, reward modeling in § 3.3, and iterative preference learning in § 3.4.

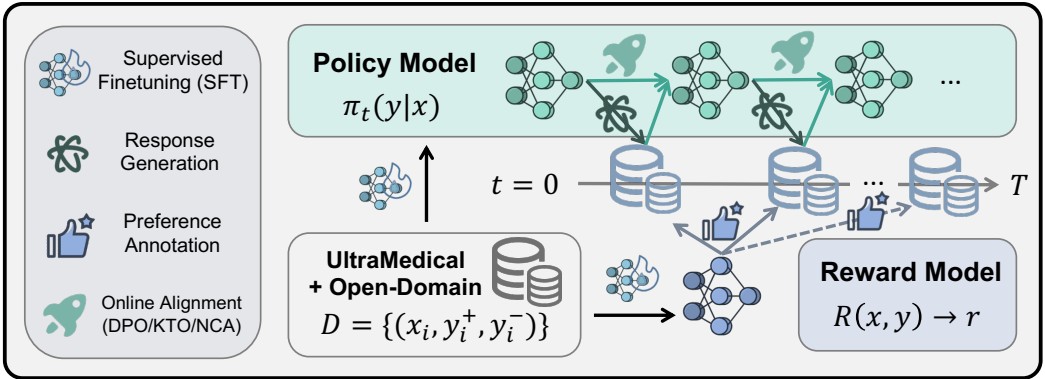

Figure 4: Process of Online Preference Learning.

## 3.1 Supervised Fine-Tuning

We conduct supervised fine-tuning (SFT) on the Llama-3 8B and 70B base models using the UltraMedical collection, resulting in `Llama-3-8B/70B-UltraMedical`. Given the uniform format of the completions, we employ responses from `gpt-4-turbo` for SFT, which consistently provide the highest quality across various sources. To enhance general instruction-following capabilities, we integrate UltraMedical with general domain datasets such as UltraChat [15], ShareGPT [57], Open-Orca [38, 45] and others. There is about 410K medical-domain and 190K open-domain samples. We retain instructions that achieve high evaluation scores in `0-hero/Matter-0.1` project[2].

## 3.2 Preference Learning

Building on the UltraMedical preferences annotation and the SFT version of UltraMedical LMs, we explore various preference learning technologies, including DPO [52] and KTO [17]. As detailed in Section 2.2.2, each instruction in UltraMedical is associated with eight completions, yielding a maximum of $C_8^2$ pairs, which is approximately 20 times the size of the instruction set used for SFT. Due to computational limitations, we utilized only the binarized version of the preference data, consisting of about 100K instructions (noted as *UltraMedPref*), where each instruction includes one chosen and one rejected response. Similarly to SFT, we incorporated the general preference datasets including UltraFeedback, UltraInteract, and UltraSafety to maintain broad capabilities, totaling approximately 75K instructions (named as *UltraMixPref*).

## 3.3 Reward Modeling

The reward model is a crucial component in technologies such as Reinforcement Learning from Human Feedback (RLHF), Rejected Sampling Fine-tuning (Interactive SFT), Iterative Direct Preference Optimization (Iterative DPO), and other continuous alignment methods. To further enhance medical language models, we train a reward model (RM) for continual alignment. The RM is trained using the preference data outlined in § 2.2.2. Besides of preference data from UltraMedical, we also augment training with UltraFeedback [13], UltraSafety [19] and UltraInteract [78] datasets to enhance its capabilities in general chat, safety, and reasoning. Subsequently, this model is employed to label responses from UltraMedical LMs and provide "on-policy" completion pairs for prefernce learning. It can also be used to evaluate numerous decoding candidates in massive sampling scenarios.

## 3.4 Iterative Preference Learning

Based on the reward model, we implement online preference learning and Best of N (BoN) sampling to further enhance the UltraMedical LMs, which can be synergistically combined to boost performance.

**Online Preference Learning:** After supervised fine-tuning on a mixture of general and medical domain instructions, we obtain the UltraMedical LM with parameters $\pi_0$. Subsequently, we conduct

---

[2] https://huggingface.co/datasets/0-hero/Matter-0.1

Table 2: Main results on medical multiple-choice questions: Models denoted with 🏥 are specifically fine-tuned using medical domain instructions. Those marked with ★ are fine-tuned with our proprietary UltraMedical dataset. Within each segment of the results, the highest scores are emphasized in **bold** and the second highest scores are indicated with underline.

| Instruct Model & Task | MedQA (US 4-opt) | MedMCQA (Dev) | PubMedQA (Reasoning) | MMLU Clinical knowledge | Medical genetics | Anatomy | Professional medicine | College biology | College medicine | Avg. |
|---|---|---|---|---|---|---|---|---|---|---|
| *~7B Models (0-shot CoT)* | | | | | | | | | | |
| Mistral-7B-Instruct* | 37.0 | 31.9 | 44.2 | 51.7 | 57.0 | 51.1 | 47.4 | 42.2 | 43.4 | 45.10 |
| Starling-LM-7B-beta* | 50.6 | 45.3 | 67.2 | 66.4 | 67.0 | 57.8 | 64.0 | 67.4 | 60.7 | 60.71 |
| 🏥 BioMistral-7B | 46.6 | 45.7 | 68.1 | 63.1 | 63.3 | 49.9 | 57.4 | 63.4 | 57.8 | 57.26 |
| 🏥 Meerkat-7B (Ens) | **74.3** | **60.7** | - | 61.9 | 70.4 | 61.5 | 69.5 | 55.4 | 57.8 | 63.94 |
| Llama-3-8B-Instruct* | 60.9 | 50.7 | 73.0 | 72.1 | 76.0 | 63.0 | 77.2 | 79.9 | 64.2 | 68.56 |
| 🏥 Internist-7B | 60.5 | 55.8 | **79.4** | 70.6 | 71.0 | 65.9 | 76.1 | - | 63.0 | 67.79 |
| 🏥 OpenBioLLM-8B | 59.0 | 56.9 | 74.1 | **76.1** | **86.1** | 69.8 | **78.2** | **84.2** | **68.0** | 72.48 |
| *★ Llama-3-8B UltraMedical (Our)* | | | | | | | | | | |
| UltraMed + SFT | 73.3 | 61.5 | 77.0 | 78.9 | 78.0 | 74.1 | 83.8 | 78.5 | 71.7 | 75.20 |
| UltraMed + Vanilla DPO | 73.7 | 63.6 | 78.2 | 76.2 | 88.0 | 75.6 | 83.8 | 79.9 | 70.5 | 76.61 |
| UltraMed + Vanilla KTO | 72.7 | 63.3 | 79.2 | 77.0 | 87.0 | 69.6 | 86.4 | 81.9 | 72.3 | 76.61 |
| UltraMix + SFT | 74.5 | 62.0 | 79.2 | 75.8 | 83.0 | 73.3 | 83.5 | 81.2 | 70.5 | 75.90 |
| UltraMix + Vanilla DPO | 74.9 | 63.6 | 79.4 | 78.1 | 84.0 | 71.9 | 86.8 | 80.6 | 76.3 | 77.29 |
| UltraMix + Vanilla KTO | 73.3 | 63.8 | 79.0 | 77.4 | 87.0 | 71.9 | 85.3 | 80.6 | 72.3 | 76.74 |
| UltraMix + Iterative DPO | 74.2 | 62.7 | 79.2 | 78.1 | 87.0 | 76.3 | 87.5 | 82.6 | 69.9 | 77.51 |
| UltraMix + Iterative KTO | 74.8 | 63.6 | 78.8 | 77.0 | 91.0 | 75.6 | 83.8 | 79.9 | 72.3 | 77.41 |
| UltraMix Best (Ens) | **76.1** | **65.3** | 79.0 | 77.7 | 87.0 | 74.8 | 87.1 | 82.6 | 75.1 | 78.32 |
| *>40B Models (0-shot CoT)* | | | | | | | | | | |
| 🏥 Med42-70B | 66.6 | 60.6 | 67.2 | 76.6 | 77.0 | 66.7 | 79.8 | 75.7 | 66.5 | 70.74 |
| Mixtral-8x7B-Instruct* | 52.8 | 49.7 | 46.2 | 71.7 | 70.0 | 62.2 | 71.0 | 77.8 | 67.1 | 63.17 |
| Mixtral-8x22B-Instruct* | 73.1 | 63.3 | 71.4 | 84.2 | 89.0 | 77.0 | 88.2 | 88.2 | 78.0 | 79.16 |
| Qwen1.5-72B-Chat* | 63.6 | 59.0 | 32.4 | 78.9 | 80.0 | 68.9 | 82.7 | 91.0 | 75.7 | 70.24 |
| Llama-2-70B-Chat* | 47.3 | 41.9 | 63.8 | 64.9 | 70.0 | 54.1 | 59.2 | 66.7 | 61.3 | 58.80 |
| Llama-3-70B-Instruct* | **79.9** | 69.6 | 75.8 | 87.2 | 93.0 | 76.3 | 88.2 | 92.4 | 81.5 | 82.66 |
| DeepSeek-v2-Chat* | 68.6 | 61.5 | 71.0 | 83.0 | 90.0 | 73.3 | 86.8 | 88.9 | 78.0 | 77.90 |
| 🏥 OpenBioLLM-70B | 78.2 | **74.0** | **79.0** | 92.9 | 93.2 | 83.9 | **93.8** | 93.8 | **85.7** | **86.06** |
| 🏥 OpenBioLLM-70B (Ens)* | 77.5 | 73.7 | **79.0** | **93.6** | **95.0** | **85.9** | 87.9 | **95.1** | 85.5 | 85.92 |
| *★ Llama-3-70B UltraMedical (Our)* | | | | | | | | | | |
| UltraMed + SFT | 82.2 | 72.3 | 78.8 | 86.4 | 91.0 | 82.2 | 92.3 | 89.6 | 86.7 | 84.62 |
| UltraMed + Vanilla DPO | 85.3 | 73.0 | 78.8 | 86.4 | 92.0 | 84.4 | 94.1 | 91.7 | 84.4 | 85.57 |
| UltraMed + Vanilla KTO | 84.7 | 73.0 | 79.8 | 86.0 | 93.0 | 84.4 | 92.6 | 93.1 | 81.5 | 85.35 |
| UltraMix + SFT | 83.7 | 73.0 | 77.6 | 84.9 | 94.9 | 80.7 | 91.9 | 91.0 | 81.5 | 84.27 |
| UltraMix + Vanilla DPO | 84.0 | 74.1 | 77.4 | 85.7 | 95.0 | 80.7 | 93.8 | 94.4 | 85.0 | 85.56 |
| UltraMix + Vanilla KTO | 84.8 | 73.2 | 80.0 | 86.8 | 92.0 | 84.4 | 93.8 | 93.1 | 84.4 | 85.84 |
| UltraMix Best (Ens) | **85.4** | **74.7** | 78.8 | 89.4 | 95.0 | 85.2 | 92.6 | 95.1 | 82.1 | 86.49 |
| *Proprietary Models (Mixed - few-shot, self-consistency)* | | | | | | | | | | |
| GPT-3.5-Trubo | 57.7 | 72.7 | 53.8 | 74.7 | 74.0 | 65.9 | 72.8 | 72.9 | 64.7 | 67.70 |
| Flan-PaLM (best) | 67.6 | 57.6 | 79.0 | 80.4 | 75.0 | 63.7 | 83.8 | 88.9 | 76.3 | 74.70 |
| GPT-4 (5-shot) | 81.4 | 72.4 | 75.2 | 86.4 | 92.0 | 80.0 | 93.8 | 95.1 | 76.9 | 83.69 |
| GPT-4 (0-shot CoT) | 85.8 | 72.3 | 70.0 | 90.2 | 94 | 84.4 | 94.5 | 93.8 | 83.2 | 85.36 |
| 🏥 Med-PaLM 2 (ER) | 85.4 | 72.3 | 75.0 | 88.7 | 92.0 | 84.4 | 92.3 | 95.8 | 83.2 | 85.46 |
| GPT-4-base (5-shot) | 86.1 | 73.7 | 80.4 | 88.7 | 97.0 | 85.2 | 93.8 | 97.2 | 80.9 | 87.00 |
| GPT-4 (Medprompt) | **90.2** | **79.1** | **82.0** | **95.8** | **98.0** | **89.6** | **95.2** | **97.9** | **89.0** | **90.76** |

inference on a mixture of instructions using $\pi_0$ and annotate the generated completions and references as "chosen" and "rejected" answers using a reward model. We then perform preference learning on the on-policy preference data, resulting in $\pi_1$. This procedure is repeated $K$ times, culminating in the final UltraMedical LM with parameters $\pi_K$.

**Best of N (BoN) Sampling:** Self-consistency is a useful method for enhancing model performance across various tasks. Previous studies, such as MedPrompt [46] and MedPaLM [59], have adapted self-consistency to achieve superior outcomes in medical QA tasks. Rather than merely voting for the majority, we employ a reward model to select the best completion from N sampling candidates. BoN sampling can be applied not only during inference but also throughout training, thereby enabling the selection of potentially better answers and refining the model's behavior.

# 4 Evaluation of UltraMedical LMs

## 4.1 Experimental Setup

**Medical domain benchmarks:** To assess the specialized capabilities of UltraMedical-based LLMs within the medical field, we evaluated these models using well-known medical question-answering benchmarks, as utilized in MedPaLM experiments. These benchmarks include MedQA [27], PubMedQA [28], MedMCQA [48], and the medical categories in MMLU [21]. We selected the

Table 3: Performance metrics of different open-source models across various general benchmarks.

| Instruct Model | K-QA | | MT-Bench | AlpacaEval 2 | | MMLU | GPQA | GSM8K |
|---|---|---|---|---|---|---|---|---|
| | Compl. (↑) | Hall. (↓) | GPT-4 | LC (%) | WR (%) | 5-shot | 0-shot | 8-shot, CoT |
| Mistral-7B-Instruct | 0.5335 | 0.2090 | 6.84 | 17.1 | 14.7 | 58.4 | 26.3 | 39.9 |
| Llama-3-8B-Instruct | 0.6037 | 0.1940 | 8.10 | 22.9 | 22.6 | 68.4 | 34.2 | 79.6 |
| OpenBioLM-8B | 0.3135 | 0.1194 | 4.38 | 0.06 | 0.25 | 44.2 | 24.8 | 41.6 |
| ★ UltraMedLM 8B | 0.7242 | 0.0945 | 7.64 | 30.7 | 31.9 | 68.1 | 34.2 | 75.9 |
| Mixtral-8x7B | 0.6617 | 0.1343 | 8.30 | 23.7 | 18.3 | 70.6 | 39.5 | 93.0 |
| Llama-3-70B-Instruct | 0.6545 | 0.1357 | 9.01 | 34.4 | 33.2 | 82.0 | 39.5 | 93.0 |
| OpenBioLM-70B | 0.5951 | 0.1100 | 8.53 | 30.8 | 31.0 | 60.1 | 29.2 | 90.5 |
| ★ UltraMedLM 70B | 0.6077 | 0.0896 | 8.54 | 33.0 | 32.1 | 77.2 | 39.7 | 88.7 |
| GPT-3.5-Turbo (1106) | 0.6208 | 0.0746 | 8.32 | 19.3 | 9.2 | 70.0 | 28.1 | 57.1 |
| GPT-4-Turbo (1106) | 0.6390 | 0.1095 | 9.32 | 50.0 | 50.0 | 86.4 | 49.1 | 92.0 |

medical categories in MMLU based on previous works, which mainly comprise Clinical Knowledge, Medical Genetics, Anatomy, Professional Medicine, College Biology, and College Medicine. In addition to these medical multiple-choice questions (MCQs), we also report results on free-form clinical questions task, named K-QA [42]. Details of these benchmarks are displayed in Appendix C.6.

**General domain benchmarks:** We evaluated the general capabilities of the models on benchmarks related to general-domain chat (MT-Bench [83] and Alpaca-Eval [37]), general MCQs (MMLU [34] and GPQA [54]), and mathematical tasks (MATH [22] and GSM8k [12]).

**Evaluation metrics:** For multiple-choice QA tasks, we use the accuracy metric. For free-form QA, we use GPT-4 as a human proxy to evaluate the results from multiple aspects. Further details about the evaluation benchmarks are available in Appendix C.6.

**Baseline Models:** We select a range of baseline models from both proprietary and open-source categories, encompassing general and medical domains. In the proprietary category, we choose GPT3.5 and GPT-4 as generalist models, and MedPaLM and MedGemini from the medical domain. In the open-source category, we include models such as Qwen [5], Mixtral [26], DeepSeek [14] and the Llama series. We also conduct comparisons with advanced medical variants, like Med42 [11], BioMistral [31], Meerkat [30], and Internist [3] and OpenBioLLM [3]. For models marked with an asterisk (*), we conduct experiments and gather results directly. Other results are adapted primarily from the literature, mainly in MedPrompt [46]. And "Ens" denotes an ensemble with 10 self-consistency responses, maintaining consistency with previous MedPrompt papers.

**Implement Details:** We apply two data settings for SFT and preference learning, where *UltraMed* only contains 410K instructions UltraMedical and *UltraMix* contains totally 600K instructions with additional 190K from general domain datasets mainly including UltraChat [15], Open-Orca [38], and EvolInstruct [72]. For preference learning, we note training on 100K *UltraMedPref* and 75K *UltraMixPref* as *Vanilla* versions, and on these instructions with annotated sampling completions as *Iterative* versions. More training details are provided in Appendix B.

## 4.2 Main Results

As shown in Table 2, the UltraMedical series, particularly the 8B models, achieve advanced performance on medical benchmarks, demonstrating the effectiveness of the UltraMedical instructions and preference datasets. To gain a deeper understanding of the results, we conducted further analyses from three perspectives: 1) the impact of incorporating open-domain instructions and preferences for Supervised Fine-Tuning (SFT) and various Preference Optimization (xPO) techniques; 2) the effectiveness of online preference learning across small and large language models (SLMs and LLMs); and 3) the trade-offs in performance between the medical and general domains.

**Dataset Mixture for SFT and xPO:** As shown in Table 2, UltraMedical LMs under the *UltraMed* settings achieve advanced performance on average scores. The models perform slightly better with the *UltraMix* datasets. This evidence supports the conclusion that a data mixture of both medical and open domains enhances both SFT and xPO processes. This also suggests that LLMs may require

---

[3]https://huggingface.co/internistai/base-7b-v0.2

Table 4: Performance of Reward Models on UltraMedical and RewardBench.

| Reward Model | UltraMedical | | | | | RewardBench | | | | |
|---|---|---|---|---|---|---|---|---|---|---|
| | Easy | Hard | Human | Length | Avg. | Chat | Chat Hard | Safety | Reasoning | Avg. |
| openbmb/UltraRM-13b | 90.34 | 73.98 | 69.33 | 66.67 | 75.08 | 96.40 | 55.50 | 56.00 | 62.40 | 67.58 |
| openbmb/Eurus-RM-7B | 89.50 | 72.96 | 73.01 | 68.33 | 75.95 | 98.04 | 62.72 | 81.89 | 89.38 | 83.01 |
| sfairXC/FsfairX-LLaMA3-RM-v0.1 | 92.86 | 70.41 | 73.62 | 67.22 | 76.03 | 99.16 | 64.69 | 86.89 | 90.64 | 85.34 |
| RLHFlow/PairRM-LLaMA3-8B | 95.80 | 72.70 | 74.85 | 70.56 | 78.48 | 98.30 | 65.80 | 89.70 | 94.70 | 87.13 |
| ★ UltraMedRM-8B | 94.12 | 73.47 | 77.30 | 77.22 | 80.53 | 97.21 | 67.11 | 91.19 | 86.62 | 85.53 |

Table 5: Comparative performance of self-consistency (SC) and reward model (RM) sorting.

| Instruct Model | SFT | | | | DPO | | | | KTO | | | |
|---|---|---|---|---|---|---|---|---|---|---|---|---|
| | Greedy | SC | UM.RM | Gen.RM | Greedy | SC | UM.RM | Gen.RM | Greedy | SC | UM.RM | Gen.RM |
| Llama3-8B-Instruct | 68.56 | 71.45 | 71.40 | 62.89 | - | - | - | - | - | - | - | - |
| Llama3-8B-UltraMed | 75.20 | 78.33 | 78.67 | 78.25 | 76.61 | 78.28 | 78.31 | 76.60 | 76.61 | 77.61 | 77.81 | 76.80 |
| Llama3-8B-UltraMix | 75.90 | 78.40 | 79.52 | 77.76 | 77.29 | 78.32 | 78.02 | 77.14 | 76.74 | 77.98 | 77.21 | 75.68 |
| Llama3-70B-Instruct | 82.66 | 83.71 | 83.74 | 81.38 | - | - | - | - | - | - | - | - |
| Llama3-70B-UltraMed | 84.62 | 86.48 | 85.61 | 85.36 | 85.57 | 86.41 | 86.27 | 85.56 | 85.35 | 86.43 | 86.18 | 85.59 |
| Llama3-70B-UltraMix | 84.27 | 86.92 | 85.30 | 85.17 | 85.56 | 86.11 | 85.62 | 85.12 | 85.84 | 86.49 | 85.84 | 85.56 |

general capabilities to solve specialized domain problems, underscoring the necessity for specialized generalists. Better mixture strategy for general and specialized data still requires exploration.

**Offline and Online Preference Learning:** The results in Table 2 indicate that the constructed preference data can enhance the performance of the 8B and 70B models through various Preference Optimization (xPO) techniques. However, the improvements are not particularly significant, especially for larger models like the 70B. The primary reasons for this lie in the differences between offline and online optimization. Although completions from advanced models are obtained, there still exists a distribution mismatch for advanced models like Llama-3. To further enhance performance, it would be beneficial to sample completions from the model itself and then apply rewards with a reward model. Further exploration of transitioning preference learning from offline to online is necessary.

**Trade-off Performance in Medical and Open Domain:** As illustrated in Table 2 and Table 3, the UltraMedical LMs benefit from a mixture of medical and general domain datasets during the Supervised Fine-Tuning (SFT) and various Preference Optimization (xPO) processes. This strategy enhances performance on medical tasks but slightly reduces results on general domain benchmarks, highlighting the potential and necessity of developing specialized generalists. This noticeable performance trade-off warrants further investigation into the principles of data mixing and its influence on downstream performance in both specialized and general tasks.

# 5 Evaluation of Reward Models

## 5.1 Setup

**Benchmark:** To assess the rewarding capabilities in the general domain, we adapted the AllenAI RewardBench, which features a variety of prompts from categories such as Chat, Chat Hard, Safety, and Reasoning. Considering that many models were trained on the prior preference dataset, we have excluded results from those prior sets in RewardBench. Furthermore, to evaluate the effectiveness of the UltraMedical reward models alongside general domain reward models in the medical domain, we conducted assessments using the UltraMedical preference dataset constructed in § 2.2.2.

**Models:** We primarily compared the performance of typical models on RewardBench, including UltraRM [13], Starling-RM [84], Eurus-RM [78], and LlaMA3-RM [16]. These reward models, along with our UltraMedical RMs, are well-suited for large-scale reward computations. Simultaneously, we also compared pairwise models like PairRM-LLaMA3 [16]. Although this model achieves high performance, it fails to scale up due to the limitations of pairwise comparison.

## 5.2 Main Results

**Performance on RewardBench:** As illustrated in Table 4, the UltraMedical RM trained sorely on Ultra-Series datasets performs competitively in both medical and general reward benchmarks. While some models exhibit strong performance on the general RewardBench, they show weaknesses in

the medical domain. The narrowing gap between models in the medical domain, compared to the general domain, suggests potential overfitting in the general domain and underscores the necessity of developing reward models specifically for the medical domain.

**Contribution to Online Preference Learning:** As demonstrated in Table 2, UltraMedical RM is effective for online/iterative preference learning methods such as DPO and KTO. Unlike the vanilla xPO settings, which utilize annotated preferences by GPT-4 and completions from multiple models, iterative xPO uses only the model's own completions, annotated by reward models. Due to computational limitations, we conducted only one round of annotation, but we plan to explore further steps like self-rewarding [79] in future work.

**Results of Re-ranking:** As shown in Table 5, reward models are not only useful for providing feedback in preference learning but also for re-ranking candidates. Our findings indicate that reward models outperform self-consistency ensembles with 8B models but are less effective in supervising 70B models, although they still facilitate preference learning. This underscores the necessity for future research to explore the re-ranking of massive candidates and the selection of the most positive ones to enhance specialized abilities, particularly focusing on weak to strong supervision [7].

**Challenges in Medical Rewarding:** In our implementations, preferences from GPT-4 are utilized to train reward models. While this AI-generated feedback is effective in the general domain, it shows some limitations in the medical domain. The UltraMedical reward benchmark indicates there is substantial room for improvement, as shown by performance on the Hard, Human, and Length sets. We plan to focus on enhancing domain-specific reward models in future work. Additionally, results in Table 5 reveal weaknesses in reward models, suggesting that the scalability of model size for reward applications [18] requires further validation.

## 6   Conclusion

In this paper, we introduce the UltraMedical datasets, comprising 410K high-quality instructions—a mix of synthetic and manual inputs—within the biomedical domain, which also includes 100K preference annotations. Utilizing the UltraMedical datasets, we conducted SFT and xPO on the Llama-3 series models, blending medical and general domain inputs. The outcomes across various medical and general domains demonstrate the superior performance of our models, validating the effectiveness of our datasets and underscoring the necessity of specialized generalists.

**Limitations and Future Directions** This paper acknowledges limitations related to using GPT-4 annotations, which may introduce bias. Instead, we could leverage powerful open-source models like Llama-70B to construct instructions using the pipeline described in the paper. Rather than directly using GPT-4's answers, we propose using only the instructions to implement self-rewarding alignment. Additionally, our work on iterative preference learning faces challenges due to limited resources, which presents an opportunity for further exploration in the future. Reward models are a critical component for the self-evolution of models; future research could focus on developing more robust reward models, utilizing our medical reward bench as a testbed. We believe the UltraMedical suites could pave new avenues in biomedicine.

**Acknowledgments and Disclosure of Funding**

**Acknowledgments** We express our gratitude to the clinical experts and graduate students majoring in biomedicine from The First Affiliated Hospital of Nanchang University. Their professional knowledge contributed invaluable suggestions towards the construction of UltraMedical, facilitated the correction of annotation errors, and provided a more robust reward benchmark. We also extend our appreciation to the open-source community for sharing dataset sources, which served as essential components of UltraMedical. Furthermore, we acknowledge the contributions of all open-source language model providers, whose efforts have significantly propelled the advancement of research in this domain.

**Disclosure of Funding** This work is supported by the National Science and Technology Major Project (2023ZD0121403), Young Elite Scientists Sponsorship Program by CAST (2023QNRC001), and National Natural Science Foundation of China (No. 62406165).

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

# A    Related Works

**Alignment of LLMs.** Since the emergence of ChatGPT, the three most critical steps of LLMs have been broadly proven to advance large language models toward sophisticated artificial intelligence, including pre-training on large-scale parameters and corpora, supervised fine-tuning (SFT) on high-quality annotations, and reinforcement learning from human feedback (RLHF) [47]. SFT has been extensively explored in recent years, leading to the emergence of numerous powerful chat and AI-assisted applications,such as Alpaca [63], UltraChat [15], and WizardLM [72]. Aligning Large Language Models (LLMs) with human or AI values has emerged as the next trend following supervised fine-tuning in the open-source community. Beyond instruction tuning, RLHF and DPO [52] techniques further improve LLMs by leveraging preference data and achieve strong performance in specialized domains. Unlike RLHF, DPO does not require a reward model, making it simpler to implement in practice. UltraFeedback [13] has become one of the most popular sources of preference data, contributing to the creation of powerful Zephyr models [66] through DPO. Various DPO variants like KTO [17], IPO [4], and CPO [73] have been proposed to advance preference learning in fields such as mathematics, coding, and reasoning.

Recent works show [74, 62] that DPO variants fail to compete with RLHF methods like Proximal Policy Optimization (PPO) [56] under identical settings. Concurrently, the focus on reward models has led researchers to explore interactive or online alignment, which has resulted in superior performance when combined with DPO variants [61, 49, 16]. This area remains under investigation, and the scaling laws concerning preference data also merit further study.

**LLMs for BioMedicine.** The powerful abilities of LLMs are increasingly promoting and advancing their applications in biomedicine community. There are two critical lines of research relevant to our work. The first research line amis to leverage integrating prompt and fine-tuning technologies with advanced proprietary models such as OpenAI's GPT-4 [1] and Google's PaLM and Gemini [64, 53]. The second one involves fine-tuning open-source LLMs using medical domain corpora and instructions, which has gradually mitigated the performance gap between open-sourced and proprietary models. In the realm of medical LLMs, the MedPaLM series [58, 59] acts as the first category to achieve over 60% accuracy on MedQA, surpassing human experts. This is achieved by employing chain-of-thought and instruction tuning based on 540B-parameter PaLM. Building upon this, MedPrompt [46] stands on the shoulder of GPT-4 to demonstrate that generalist LLMs can outperform medical-specific fine-tuned models by 90% on MedQA by exploiting dynamic few-shot in-context examples [6] and chain-of-thought [69] techniques, which is the first milestone model with the excellent specialized performance. MedGenimi [55] integrates web search into the loop to foster self-evolving learning in LLMs, achieving new state-of-the-art (SoTA) performance on multimodal medical benchmarks. However, these models are still closed-source and face privacy and transparency challenges in real-world applications. In the second line of development, researchers have conducted further pre-training and instruction tuning [9, 71, 20, 31, 11] on open-sourced LLMs such as Llama [65, 2] and Mistral [25, 26].

Though achieving remarkable success, open-source models still lag behind proprietary models in medical benchmarks and applications and suffer from reduced performance in general domains due to potential overfitting on medical data. Moreover, the explorations of advanced alignment technologies such as DPO, KTO, and RLHF are still limited by resource constraints in high-quality instructions and preference data. In this paper, we explore enhancement strategies to improve the medical performance of open-source models while preserving their general capabilities from data perspective and advanced alignment technologies.

# B    Training Details

**Supervised Fine-Tuning:** To preserve the general capabilities of fine-tuned models, we conducted continuous fine-tuning on instructed models for two epochs, using a learning rate of 2e-5 and a warm-up ratio of 0.1 with a cosine scheduler. For both the 8B and 70B models, we combined datasets

including 58K from UltraChat [4], 40K from Evol-Instruct-v2 [5], 30K from Open-Orca [6], 47K from Camel Instructions [7], and 16K from Orca-Math problems [8]. The maximum length is set to 2048.

**Preference Learning:** For hyper-parameters of DPO and KTO, we explore learning rates of {1e-7, 3e-7, 5e-7} and $\beta$ values of {0.01, 0.05, 0.1, 0.4}. Each model is fine-tuned for one epoch with a warmup ratio of 0.1 using a cosine scheduler. We utiliz the implementations from the trl library [9] and employ the *kto-pair* loss for KTO training. The maximum length is set to 2048.

**Reward Modeling:** We train reward models for 1 epoch using a learning rate of 2e-6 on the Llama3-8B-Instruct model, employing a cosine scheduler with a warmup ratio of 0.1. The maximum length for instruction and response is set to 2048.

**Iterative Preference Learning:** For 100K instructions in UltraMedical, we generate five candidate responses for each instruction. We use a sampling strategy with a temperature of 0.8 for decoding. Each response is then annotated with the reward model and sorted from highest to lowest. For QA problems with a golden choice, the highest-reward correct response is selected as "chosen," and the lowest-reward incorrect response as "rejected," in a strategy known as rejected sampling. For open-ended instructions, the responses with the highest and lowest rewards are directly selected as "chosen" and "rejected," respectively. We conduct xPO on SFT models using the rewarding preference for 1 epoch and optimize the hyper-parameters consistent with those used above.

## C   Dataset Details

### C.1   Details of UltraMedical Instructions

We display the composition of the UltraMedical collections in Figure 5a, where multi-choice question answering comprises about 50%, PubMed question answering accounts for about 20%, and the remaining 30% consists of open-ended instructions and dialogues. As displayed in Figure 3, we randomly selected 200K prompts from the UltraMedical collection and mapped them into vectors using Atlas Nomic.AI. We present the topic distribution in Figure 3 and the task distribution in Figure 5b, both of which validate the effectiveness of our diversity-driven process. Details about the map can be viewed through this Nomic AI Atlas.

### C.2   Details of UltraMedical Preference

We present the model's accuracy for QA tasks in Figure 9a, the models' win percentages in binarized preference in Figure 12, and the scores and rankings of all models across various tasks from GPT-4 in Figures 11 and 10.

### C.3   Details of Medical Reward Bench

For the easy set, we selected `gpt-4-1106-preview` as the chosen model, while `gpt-3.5-turbo-1106`, `Mixtral-8x22B-Instruct`, and `Mixtral-8x7B-Instruct` were rejected. For the hard set, we selected models with the highest and nearly highest scores, including `gpt-4-1106-preview`, `Meta-Llama-3-70B-Instruct`, and `Llama-3-8B-UltraMedical`. For the set without length bias, we selected `Meta-Llama-3-70B-Instruct` and `Meta-Llama-3-8B-Instruct` as chosen and rejected, respectively, which have a significant gap in scores but the same answer length.

For the initially given 1,000 test pairs, we ultimately retained 777 pairs following human expert annotation. These include 238 easy, 196 hard, 180 length-based, and 163 human-judged pairs. Approximately 233 pairs were filtered out due to issues such as incorrect formulations, difficulty in answering, or both. The human category comprises pairs where preferences differ between human annotators and GPT-4, which is regraded as even hard for GPT-4 to recognize.

---

[4]https://huggingface.co/datasets/stingning/ultrachat
[5]https://github.com/nlpxucan/WizardLM
[6]https://huggingface.co/datasets/Open-Orca/OpenOrca
[7]https://huggingface.co/camel-ai
[8]https://huggingface.co/datasets/microsoft/orca-math-word-problems-200k
[9]https://github.com/huggingface/trl

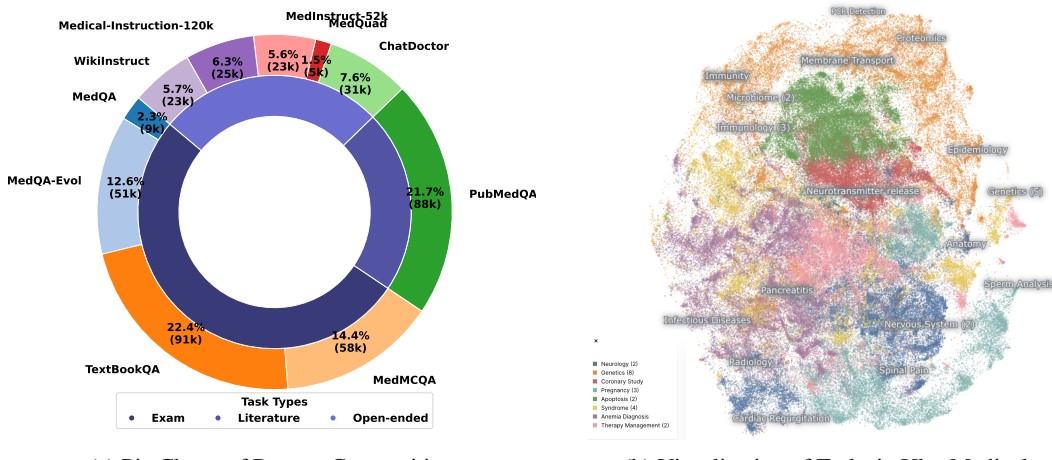

(a) Pie Charts of Dataset Composition      (b) Visualization of Tasks in UltraMedical

Figure 5: Statistics of UltraMedical dataset.

## C.4 Details of Human Annotation

We requested a human expert to review and re-annotate 1,000 preference pairs. The web interface used for annotation is displayed in Figure 6.

For biomedical-related questions (including clinical, exam, and open research questions) along with responses from Model A and Model B, the task is to choose the best response (vote) based on the following criteria:

- "Honest": The response is more accurate and verifiable with no factual errors.
- "Helpful": The response is more useful, addresses the problem effectively, and does not contain platitudes.
- "Harmless": The response is safe and ethically sound (generally always the case).
- "Length Bias": A longer response is not necessarily better; avoid verbosity, prioritizing the above 3H.

Annotation Process:

- Enter the name of the annotator for later processing and filtering of invalid annotations.
- Review the question and the answers from the two models (for reference, see GPT-4's explanation), and vote for the best response.
- After clicking to vote, the question will automatically refresh. To simplify the process, returning to a previous question is not supported!
- If a suitable response cannot be selected due to a lack of knowledge, errors in the question or answers, etc., click skip.
- Also, leave a comment in the comments section with the corresponding question number.

## C.5 Details of General Instructions

To enhance the general instruction-following capabilities, we integrate the UltraMedical with high-quality prompts from various general domains, sourced from UltraChat, Dolphin, Wizard, Orca, and additional datasets included in `0-hero/Matter-0.1`.

## C.6 Details of Benchmarks

The number of evaluations and descriptions of the tasks are presented in Table 7.

Figure 6: The WebUI used by human experts to annotate and correct preferences.

Table 6: Statistics of datasets in UltraMedical.

| Type | Dataset | Count | Description |
|------|---------|-------|-------------|
| Exam | MedQA | 9,343 | General medical knowledge in US medical licensing exam |
| | MedMCQA | 58,871 | General medical knowledge in Indian medical entrance exams |
| | MedQA-Evol | 51,809 | Synthesized data evolved from the original MedQA data |
| | TextBookQA | 91,684 | Multiple-choice questions derived from medical books |
| Literature | PubMedQA | 88,688 | Closed-domain question answering given PubMed abstract |
| Open-End | MedQuad | 5,957 | Medical question-answer pairs created from 12 NIH websites |
| | MedInstruct-52k | 23,032 | Generated medical instruction-following data with self-instruct |
| | Medical-Instruction-120k | 25,806 | Various thoughts proposed by the people and synthetic responses |
| | ChatDoctor | 31,115 | Real conversations between patients and doctors from HealthCareMagic |
| | WikiInstruct | 23,288 | Detailed knowledge and instructions expanded from thousands of biomedical concepts from Wikipedia pages. |

# D  Dataset Analysis

## D.1  Correlation of model-based scores

We have selected `gpt-3.5-turbo` as the evaluator for instruction scoring, as it remains highly competitive with mainstream open-source LLMs and offers scalability due to its lower cost. `gpt-3.5-turbo` demonstrates a high correlation and maintains stability across multiple evaluation iterations, as shown on the left side of Figure 7. Additionally, `gpt-3.5-turbo` exhibits a strong correlation with `gpt-4-turbo`, as depicted in the middle of Figure 7. The primary difference is that instructions typically receive slightly lower scores in `gpt-4-turbo` evaluations.

Table 7: Statistics of datasets for evaluations.

| Domain | Dataset | Count | Description |
|---|---|---|---|
| Medical | MedQA (UCMLE) | 1273 | General medical knowledge in US medical licensing exam |
| | MedMCQA | 4183 | General medical knowledge in Indian medical entrance exams |
| | PubMedQA | 500 | Closed-domain question answering given PubMed abstract |
| | MMLU-Clinical knowledge | 265 | Clinical knowledge multiple-choice questions |
| | MMLU-Medical genetics | 100 | Medical genetics multiple-choice questions |
| | MMLU-Anatomy | 135 | Anatomy multiple-choice questions |
| | MMLU-Professional medicine | 272 | Professional medicine multiple-choice questions |
| | MMLU-College biology | 144 | College biology multiple-choice questions |
| | MMLU-College medicine | 173 | College medicine multiple-choice questions |
| | K-QA | 201 | Real-world clinical questions with physician-curated answers (long-form answers) |
| | MultimedQA | 140 | Consumer medical question-answering data (long-form answers) |
| General | MT-Bench | 80 | Multi-turn question answering benchmark evaluating eight different abilities |
| | Alpaca-Eval 2 | 805 | General world knowledge question-answering for chat-models |
| | Arena-Hard | 500 | Built from live data in the Chatbot Arena with challenging user queries |
| | MMLU | 116k | Multi-choice questions for massive multitask language understanding |
| | GPQA | 198 | Very hard multiple-choice and question answering tasks in biology, physics, and chemistry |
| | GSM8K | 1319 | Grade school math word problems for question answering |
| | MATH | 5000 | Challenging competition mathematics problems |

Beyond model-based scoring, previous studies have also attempted to rank instructions directly based on length. As illustrated on the right side of Figure 7, the correlation between model-based scores and lengths is very low, indicating that the evaluator prioritizes assessing instruction complexity rather than merely its length.

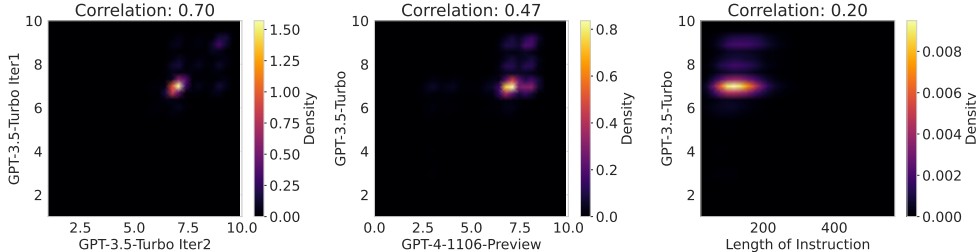

Figure 7: Correlation analysis of various scores, including those from different models and the length of instructions.

## D.2 Complexity Evoling of Instructions

Complexity of instructions is a principal characteristic of high quality. For our synthetic datasets, we conduct two additional rounds of instruction evolution to increase complexity. As shown in Figure 8, the scores of instructions across the three datasets consistently increase. Within these datasets, instructions in TextBookQA are synthesized based on few-shot examples and paragraphs from textbooks, resulting in minor score changes. The WikiInstruct dataset, which includes various open-ended questions based on entities from Wikipedia, exhibits the highest complexity scores.

## D.3 Instruction Distribution

The UltraMedical collections contain three main task types and ten sub-tasks, as illustrated in Figure 5a. Questions derived from exams and textbooks account for approximately 50%, literature-based questions for about 20%, and open-ended instructions and questions for around 30%. We randomly sample 5,000 examples from each sub-task, embed them using `intfloat/e5-mistral-7b-instruct`[67], and subsequently project them into two dimensions with t-SNE. As depicted in Figure 5b, questions in the exam series exhibit broad and diverse topics, while instructions from literature and our synthetic instructions based on Wikipedia entities are complementary.

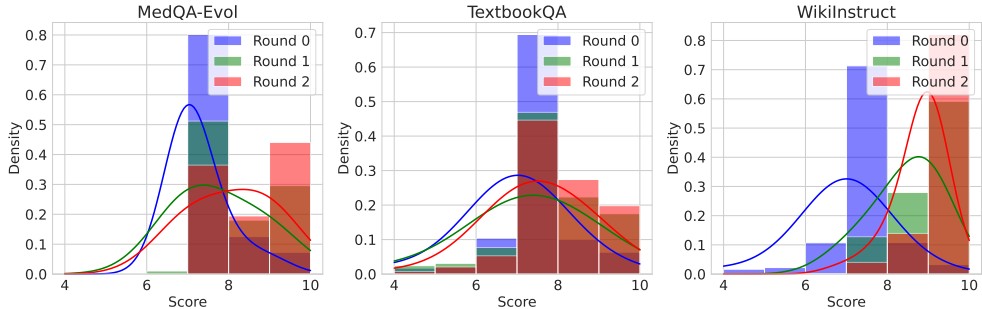

Figure 8: Distribution of model-based evaluation score progression across evolution rounds for our three synthetic datasets, illustrating how instruction evolution contributes to increased complexity.

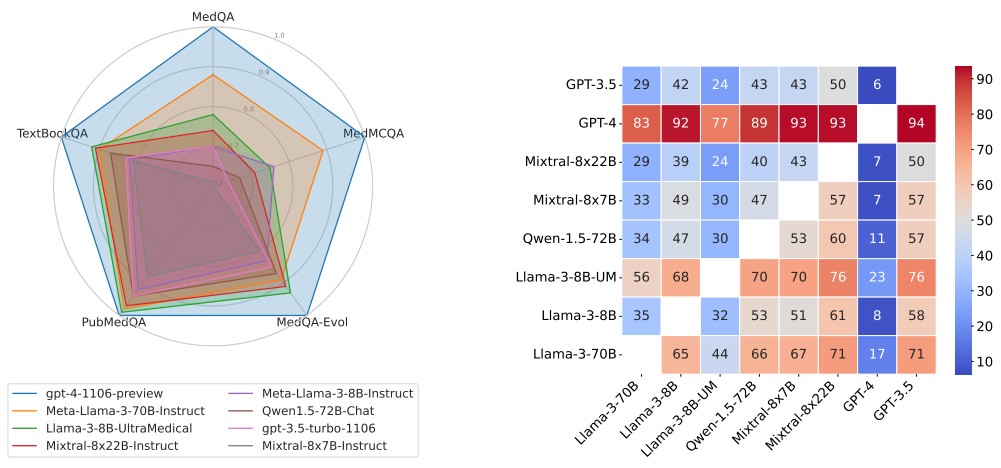

(a) Models' accuracy on QA tasks.      (b) Models's Win Percentage in Binarized Preference.

Figure 9: Annotation and preference of models statistic results.

## D.4 Instruction Decontaminate

Due to the use of large-scale synthetic data, we implement decontamination operations to prevent test set leakage, as described in the `bagel` project[10]. Our approach involves clustering all training and test data based on dense vectors and then calculating the length correlation within the top K nearest samples. This method revealed no potential test data leakage in UltraMedical.

## E UltraMedical Examples

As shown in Table 8, we provide examples from our synthetic dataset, including MedQA-Evol, TextBookQA, and WikiInstruct. For each example, we display the question, the answer provided by GPT-4, and the evaluation score on the question from GPT-3.5. Additional examples for other tasks are available on the original website, as indicated in Table 6.

Table 8: This table illustrates examples in MedQA-Evol, TextBookQA, and WikiInstruct.

---

**MEDQA-EVOL TASK**
**Question:**

---

[10]https://github.com/jondurbin/bagel/tree/main

In a 23-year-old pregnant female at 22 weeks gestation with a confirmed urinary tract infection experiencing dysuria, which condition could potentially develop as a direct sequela of this infection if left untreated, and is particularly associated with the immunological adaptations during pregnancy?

A. Gestational diabetes
B. Preterm labor
C. Intrauterine growth restriction (IUGR)
D. Hyperemesis gravidarum

**GPT-4 Answer:**
In pregnancy, a woman's immune system undergoes various adaptations, making her more susceptible to certain infections, including urinary tract infections (UTIs). If a UTI during pregnancy is left untreated, it can ascend to the kidneys, leading to a more severe infection known as pyelonephritis. This condition can further complicate the pregnancy.

A significant risk associated with untreated UTIs in pregnancy is preterm labor. The inflammation and infection can stimulate the production of prostaglandins, which are chemicals that can induce labor. This can result in preterm birth, where the baby is born before 37 weeks of gestation.

Preterm labor is a well-recognized complication of untreated UTIs during pregnancy and is distinguished from the other options because:
- Gestational diabetes is related to hormonal changes affecting insulin, not directly linked to urinary tract infections.
- Intrauterine growth restriction (IUGR) refers to poor growth of a fetus while in the mother's womb during pregnancy and is not typically a direct consequence of UTIs.
- Hyperemesis gravidarum, characterized by severe nausea and vomiting, is not directly caused by UTIs.

So, the answer is B.

**GPT-3.5 Score:**
Score: 7

Justification: This question requires a deep understanding of both urinary tract infections and the immunological adaptations during pregnancy. It also involves understanding potential complications of untreated infections during pregnancy. The question is complex and requires a high level of medical knowledge.

## TEXTBOOKQA TASK
**Question:**
In a comprehensive regional anatomical analysis of the thorax emphasizing the respiratory system's components, which structure would NOT be expected to be scrutinized, considering the respiratory focus?

A. External intercostal muscles
B. Costovertebral joints
C. Visceral pleura
D. Coronary vessels

**GPT-4 Answer:**
The thorax, or chest region, contains several structures, including those related to the respiratory system. When focusing on the respiratory components, one would typically examine structures directly involved in the process of breathing and gas exchange.

A. External intercostal muscles - These muscles are significant in the respiratory system because they are involved in the mechanics of breathing, particularly in the elevation of the ribs during inspiration.

B. Costovertebral joints - These joints are where the ribs articulate with the vertebrae. They are relevant to the respiratory system because they allow the movement of the ribs during respiration.

C. Visceral pleura - This is the membrane that directly covers the lungs. It is definitely a part of the respiratory system, playing a role in the lubrication and protection of the lungs.

D. Coronary vessels - These are the blood vessels of the heart (coronary arteries and veins) that provide the myocardium (heart muscle) with oxygen and nutrients. They are not directly involved in the respiratory system but are part of the circulatory system.

Given that the coronary vessels are not components of the respiratory system but rather of the circulatory system, they would not be expected to be scrutinized in a comprehensive regional anatomical analysis of the thorax emphasizing the respiratory system's components.

So, the answer is D.

**GPT-3.5 Score:**
Score: 7

Justification: This question requires a deep understanding of regional anatomy of the thorax, specifically focusing on the respiratory system. It involves knowledge of anatomical structures, their relationships, and their functions. The question also requires the ability to identify which structure is not directly related to the respiratory system, adding a level of complexity.

---

### WikiInstruct Task
**Question:**
Evaluate the survival curve graph illustrating the differential survival rates of organ transplants when augmented with regulatory T cell therapy compared to a control group lacking this immunomodulatory intervention, including an analysis of the variance in outcomes. Delve into the prognostic significance of these findings for future graft viability, considering underlying immune response mechanisms.

**GPT-4 Answer:**
Evaluating a survival curve that illustrates the differential survival rates of organ transplants augmented with regulatory T cell therapy compared to a control group without this intervention requires considering several key aspects:

1. **Survival Curve Analysis**: The survival curve, often plotted as a Kaplan-Meier curve, would show the proportion of grafts surviving over time for both groups. A higher curve for the group receiving regulatory T cell therapy would indicate better survival rates. It's important to look at the separation between the two curves; a significant gap suggests a meaningful impact of the therapy.

2. **Statistical Significance**: To analyze variance in outcomes, the log-rank test or Cox proportional hazards model might be used to determine if the differences in survival rates are statistically significant. This analysis would help confirm whether the observed differences are likely due to the therapy rather than chance.

3. **Prognostic Significance**: If the survival curve demonstrates significantly better outcomes with regulatory T cell therapy, this has profound implications for future graft viability. It suggests that modulating the immune response via regulatory T cells can effectively prolong graft survival, potentially reducing the need for long-term immunosuppression, which has its own side effects.

4. **Immune Response Mechanisms**: The underlying mechanism involves regulatory T cells' role in maintaining immune tolerance. By augmenting with regulatory T cell therapy, the immune system's attack on the transplanted organ can be minimized. This therapeutic approach targets the immune system's fundamental response to foreign antigens, aiming to establish a more tolerant environment for the graft.

5. **Future Implications**: The success of regulatory T cell therapy in improving graft survival rates could revolutionize transplant medicine by offering a more targeted approach to prevent graft rejection. This could lead to protocols that improve the quality of life for transplant recipients, reduce the incidence of graft-versus-host disease, and possibly decrease the reliance on general immunosuppressants, which have broad and often detrimental effects on the patient's overall immune function.

In summary, a survival curve showing improved outcomes with regulatory T cell therapy would highlight the therapy's potential as a game-changer in transplant medicine. It underlines the importance of immune modulation in transplant success and paves the way for research into similar strategies that could further improve graft survival and patient outcomes.

**GPT-3.5 Score:**
I would rate this question a 9. The question requires a deep understanding of organ transplantation, immunology, and statistical analysis. It involves a high level of technical terminology and requires a comprehensive understanding of immune response mechanisms and their impact on graft viability.

# F   Prompts

**QA Prompts:** We use the prompts and question-answer format referenced in MedPrompt [46], where we regard all the questions in PubMedQA [28] as multiple-choice problems, including options for "yes," "no," or "maybe." We use the same zero-shot CoT prompts for all models.

**Prompts for Instruction Evaluation:** We utilize GPT-3.5 to assess the complexity of each question across all datasets. Each instruction is rated on a scale from 1 to 10, where 1 represents the easiest and 10 denotes the most challenging, as described in [39].

**Prompts for Preference Annotation:** We employ GPT-4 to annotate preferences for all completions of instructions. For each annotation, we provide GPT-4 with all candidate model responses alongside the question and potential answers (primarily for multiple-choice QA), and then instruct GPT-4 to score each response on a scale from 1 to 5, where 1 is the worst and 5 is the best, based on a 5-level requirement system. Finally, GPT-4 ranks all models according to these scores. Our approach mainly references [36] to define the 5-level requirements from a biomedicine perspective.

**Prompts for Instruction Evaluation:** We conduct instruction evaluation on MedQA problems using GPT-4. The goal of this evaluation is to enhance the complexity of the questions using four base methods, as utilized in EvolInstruct [72, 41].

**Prompts for TextBook Question Generation:** We present three examples and a paragraph from a collection of 18 widely used medical textbooks, which serve as crucial references for students preparing for the United States Medical Licensing Examination (USMLE). These textbooks can be accessed at `MedRAG/textbooks`[11].

**Prompts for Wikipedia Instruction Generation:** The process begins by crawling all topics from the BioMedicine page on Wikipedia, followed by prompting GPT-4 to generate sub-topics within this field. Subsequently, we instruct GPT-4 to create open-domain instructions for various applications, based on these sub-topics and a background introduction, akin to the approach in Self-Instruct [68].

We provide all above prompts in Table 9.

Table 9: This table displays the prompts used in our experiments.

---

**ZERO-SHOT PROMPTS FOR QA**
## Question
{{ question }}
## Task
Answer the above question with format 'So, the answer is' after your explanation. For example, if the answer is A, write 'So, the answer is A'.

---

[11] https://huggingface.co/datasets/MedRAG/textbooks

## Answer
Let's think step by step.

---

Please evaluate the following question and rate its difficulty and complexity on a scale from 1 to 10, with 1 being the least difficult/complex and 10 being the most difficult/complex. Consider factors such as the breadth and depth of knowledge required, the number of concepts involved, the level of technical terminology, and the presence of quantitative or analytical components.

In addition to the numerical score, provide a brief justification (1-2 sentences) explaining your rationale for the assigned score. This will help us better understand the reasoning behind your evaluation.

## Question
{question}

## Evaluation
Justification:
Score: [1-10]

---

Please evaluate the following user instruction and the proposed response within the context of biomedicine.

## Evaluation Criteria
Use the following 5-point scale to assess how well the AI Assistant's response addresses the biomedical inquiry:

1: Inadequate - The response is incomplete, vague, off-topic, or controversial. It may lack necessary biomedical data, use incorrect terminology, or include irrelevant clinical examples. The perspective may be inappropriate, such as personal experiences from non-scientific blogs or resembling a forum answer, which is unsuitable given the precision required in biomedicine.

2: Partially Adequate - The response addresses most biomedical aspects requested but lacks direct engagement with the core scientific question. It might provide a general overview instead of detailed biomedical mechanisms or specific clinical applications.

3: Acceptable - The response is helpful, covering all basic biomedical queries. However, it may not adopt an AI Assistant's typical scientific voice, resembling content from general health blogs or web pages and could include personal opinions or generic information.

4: Good - The response is clearly from an AI Assistant, accurately focusing on the biomedical instruction. It is complete, clear, and comprehensive, presented in a clinically appropriate tone. Minor improvements could include adding more precise scientific details or a more formal presentation.

5: Excellent - The response perfectly represents an AI Assistant in biomedicine, addressing the user's scientific inquiry without any irrelevant content. It demonstrates in-depth knowledge, is scientifically accurate, logically structured, engaging, insightful, and impeccably written.

## Question and Reference Answer
Question: {question}

Reference Answer: {answer}

## Model Responses
{candidates}

## Feedback and Rankings
Provide feedback and an overall score between 1 to 5 for each response based on the **Evaluation Criteria**. Then rank the model responses, even if they share the same score, based on criteria such as clarity of response logic, richness of information, and naturalness of language.

Format your feedback and rankings as follows:

```
{{
    "feedback": {{
    "Model 1": {{
        "Evaluation": "",
        "Score": ""
    }},
    // Similar entries for other models
}},
    "ranking": [
        {{"rank": 1, "model": "Model X"}},
        // Subsequent rankings
    ]
}}
```

---

**PROMPTS FOR INSTRUCTIONS EVOLUTION BY GPT-4**

Act as a Question Rewriter to make biomedical multiple-choice questions more challenging for AI systems like ChatGPT and GPT-4, while remaining reasonable for human experts to understand and answer.

Complicate the given question using one of these methods:

[METHOD 1] Add one more constraint or requirement.
[METHOD 2] Replace general concepts with more specific ones.
[METHOD 3] Make the choices hard to differentiate by adding more complex distractors.
[METHOD 4] If solvable with simple thinking, request multi-step reasoning.

Limit additions to 10-20 words. Ensure a unique answer exists among the choices.

Question:
{question}

Output JSON format:
``` {{
    "question": "Rewritten question in the format: "xxx\nA. xxx\nB. xxx\nC. xxx\nD. xxx"",
    "answer": "A/B/C/D"
}}
```

---

**PROMPTS FOR TEXTBOOK QUESTION GENERATION BY GPT-4**

## Paragraph from the medical textbook
{paragraph}

## Example multi-choice questions
### Example 1
Question: {example1}
Answer: {answer1}

### Example 2
Question: {example2}
Answer: {answer2}

### Example 3
Question: {example3}
Answer: {answer3}

## Instructions
1. Evaluate the examination significance of the provided paragraph.
2. Assess whether the paragraph contains sufficient knowledge to evaluate a powerful AI like GPT-4. Consider factors such as:
   - Depth and breadth of the medical concepts covered
   - Specificity and technicality of the information provided
   - Potential for testing higher-order thinking skills
3. If the paragraph is deemed significant and contains enough knowledge to evaluate GPT-4, generate a synthetic multi-choice question based on the paragraph's content and the provided examples. Ensure that the generated question has a single, unambiguous correct answer among the provided choices.
4. If the paragraph is not significant or lacks sufficient knowledge for AI evaluation, set the value of "generated_question" to an empty object ({{}}).
5. Provide the output in the specified JSON format.

## Output Format (JSON)
```
{{
  "examination_significance": boolean,
  "sufficient_knowledge_for_ai_evaluation": boolean,
  "generated_question": {{      "question": string,
    "answer_choices": [
      {{
        "choice": string,
        "correct": boolean
      }},
      {{
        "choice": string,
        "correct": boolean
      }},
      {{
        "choice": string,
        "correct": boolean
      }},
        "choice": string,
        "correct": boolean
      }}
    ]
    }}
}}
```

---

**PROMPTS FOR WIKIPEDIA SUB-TOPICS GENERATION BY GPT-4**
{entity}: {description} As an expert in the field of {entity}, I need you to do the following: 1. List {number} subfields within the realm of {entity} research.
2. Ensure that these subfields represent distinct areas of entity without any overlap.
3. Provide a brief description for each subfield, highlighting its main research focus and characteristics.
4. Aim for this list to comprehensively reflect the diversity and breadth of the biomedical field.
5. Present this list in an array of dictionaries format, with each dictionary containing two keys: 'name' (the name of the subfield) and 'description' (a brief description of the subfield).

Example output format:
```

```
   {{"name": "Gene Editing", "description": "Gene editing involves altering the genetic material
of organisms to study gene functions or treat genetic diseases."}},
   {{"name": "Neuroscience", "description": "Neuroscience focuses on the study of the structure,
function, and diseases of the nervous system."}},
   // ... 18 more subfields
```

---

**PROMPTS FOR WIKIPEDIA INSTRUCTIONS GENERATION BY GPT-4**

{topic}: {description} As an expert in the field of {topic}, please devise {number} {topic}-related questions or instructions, formatted as an array of dictionaries, each with two keys: 'instruction' and 'context'. Follow these guidelines:

1. **Verb Diversity**: Incorporate a broad spectrum of verbs to diversify and enrich the instructions set.

2. **Language Style Variability**: Blend both interrogative and imperative sentence structures to enhance the dynamism of instructions.

3. **Range of Task Types**: Ensure the tasks span a variety of categories such as explanations, analyses, comparisons, and more. 1. **Difficulty levels should vary from elementary concepts to complex scientific inquiries and extend to addressing novel, challenging scenarios.

4. **Exclusivity to Text-Based Tasks**: Frame all instructions in a text-only format. Refrain from incorporating tasks that require physical execution or laboratory experimentation.

5. **Conciseness and Precision**: Articulate each instruction in English with utmost precision, limiting it to 1 or 2 sentences for clarity and brevity.

6. **Background Information Accuracy**: For tasks necessitating supplementary context, provide succinct yet comprehensive descriptions (restricted to 100 words). For basic queries, simply state "None" in the context section.

7. **JSON Format Adherence**: Format the output as an array of dictionaries. Each dictionary should have two keys: 'instruction' for the task description and 'context' for the relevant background information.

Example output format:
```
   {{"instruction": "Explain the structure of liposomes and their role in drug delivery.", "context":
"Liposomes are nanoscale carriers used in drug delivery, where their structure and function
significantly impact efficiency."}},
   {{"instruction": "List three common cardiovascular diseases.", "context": "None"}},
   // ... 18 more instructions
```

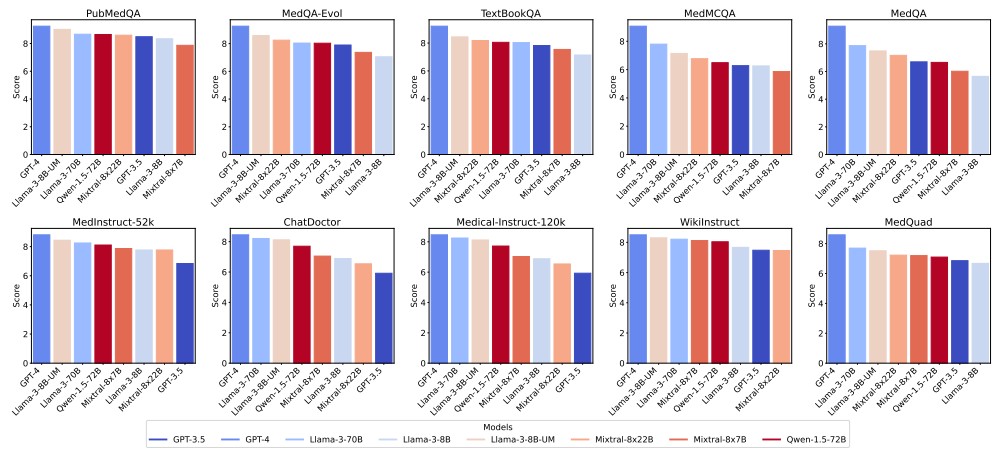

Figure 10: Scores of all models across various tasks from GPT-4 (higher is better).

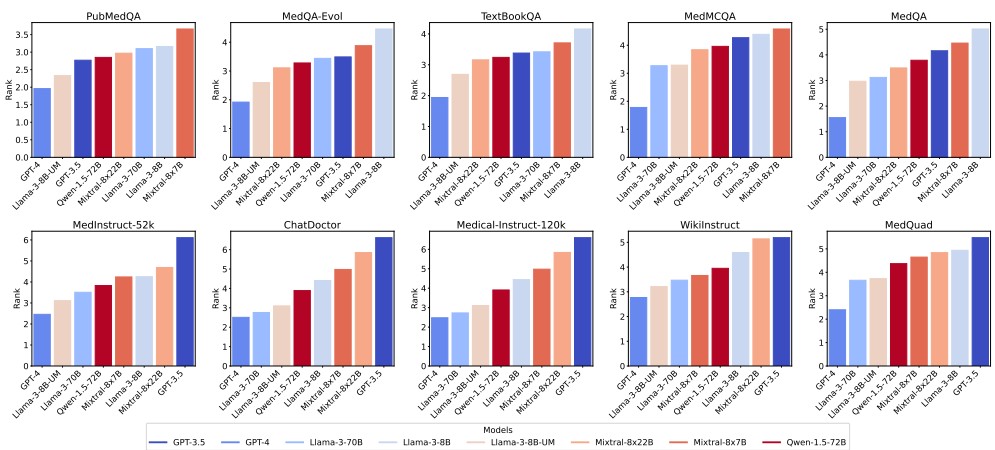

Figure 11: Ranking of all models across various tasks from GPT-4 (lower is better).

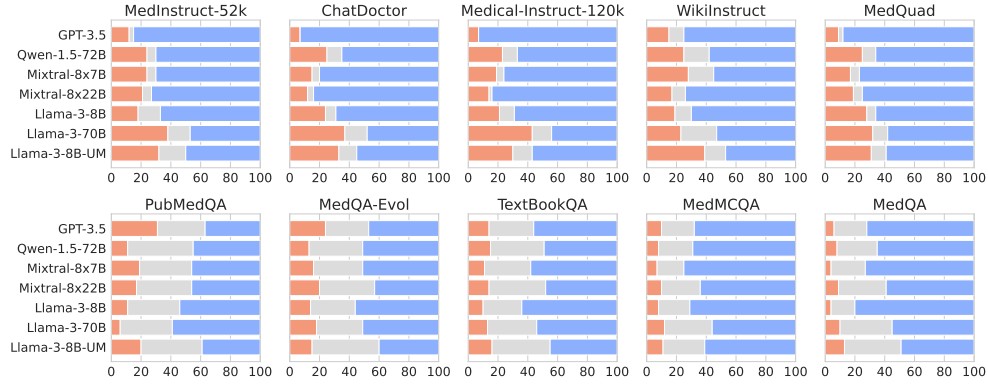

Figure 12: Ranking of models relative to GPT-4 (win/tie/loss) across various tasks, based on feedback from GPT-4. Green, gray, and red represent win, tie, and loss, respectively.

