# Supplementary Material for UltraMedical

## Contents

# 1 Dataset Details

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

## 3 Dataset Release

### 3.1 Website

We have released the datasets and models on GitHub and Huggingface, which ensure long-term availability.

GitHub: https://github.com/TsinghuaC3I/UltraMedical

Huggingface: https://huggingface.co/TsinghuaC3I

---

[1] https://github.com/jondurbin/bagel/tree/main

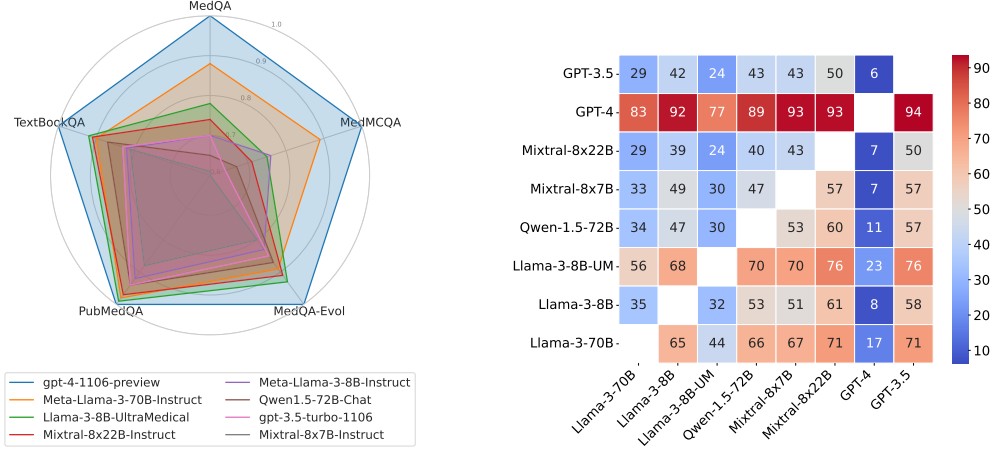

(a) Models' accuracy on QA tasks.

(b) Models's Win Percentage in Binarized Preference.

Figure 4: Annotation and preference of models statistic results.

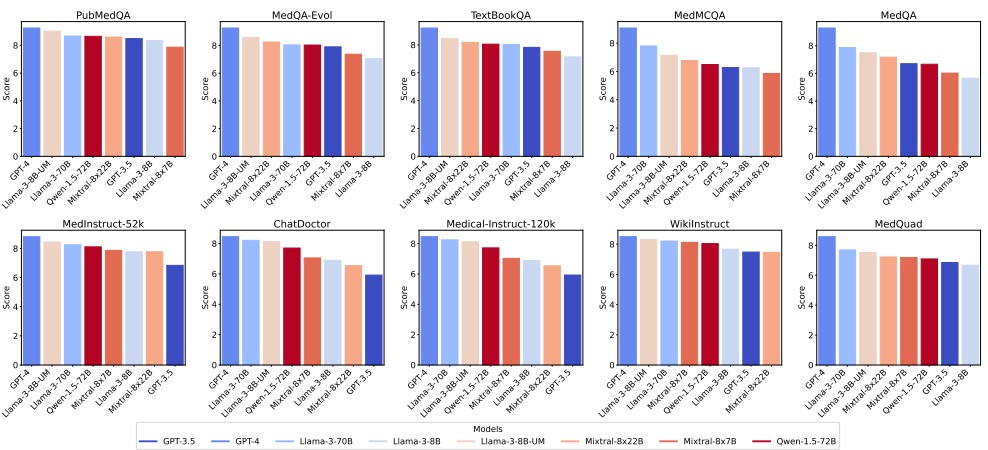

Figure 5: Scores of all models across various tasks from GPT-4 (higher is better).

We have also sampled 200,000 examples and displayed them on the Nomic AI Atlas for topic and task distribution exploration.

All model usage must adhere to the Llama3 License [2], and datasets derived from OpenAI GPT are not permitted for commercial use.

## 3.2 Limitations

While the UltraMedical suites show promising performance on several benchmarks, they still have limitations, such as hallucinations. Additionally, the outputs are synthesized from GPT-4, which may also exhibit bias. We plan to address these issues and verify the accuracy of facts in UltraMedical in future research.

---

[2] https://llama.meta.com/llama3/license/

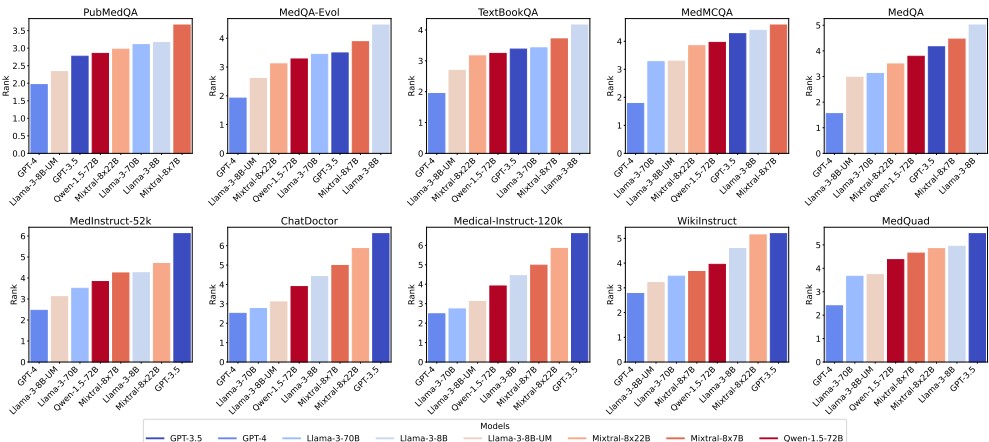

Figure 6: Ranking of all models across various tasks from GPT-4 (lower is better).

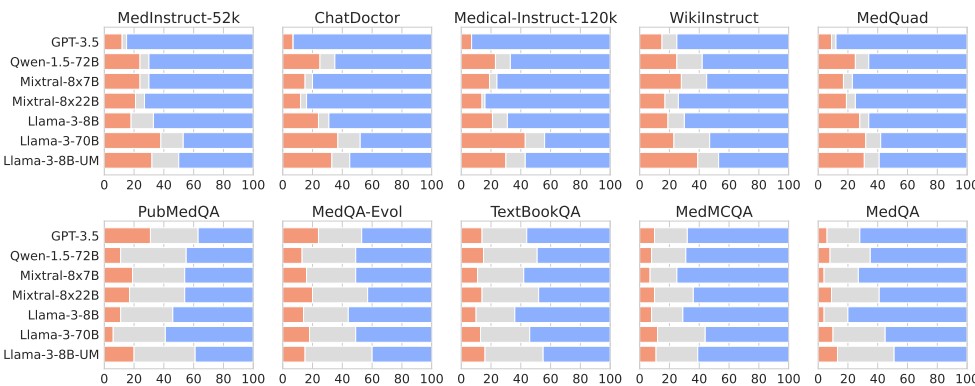

Figure 7: Ranking of models relative to GPT-4 (win/tie/loss) across various tasks, based on feedback from GPT-4. Green, gray, and red represent win, tie, and loss, respectively.

# 4 The Medical RewardBench

## 4.1 Construction

As we introduce in the main content, we firstly prompt GPT-4 for preference annotation of about 100K instructions with about 800K completions, which produce about 1.8 million (instruction, chosen, rejected) pairs. To control different level of gap between chosen and rejected completions, we further sample pairs based on easy, hard and length criteria from the 1.8 million pairs, resulting into about 100K pairs. We chose a subset pairs about 1,000 from the final 100K pairs as test set, where includes 100 for each sub-task.

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

```

## 6 Examples

We provide examples for each sub-task within UltraMedical in Table 4, including the dataset name, question, answer from GPT-4 and the original dataset, as well as the score on the instruction by ChatGPT. For the WikiInstruct task, we construct the instructions from scratch, resulting in no answer provided.

Table 4: Examples of datasets in UltraMedical.

| Dataset | Question | GPT4-Answer | Answer | Score |
|---------|----------|-------------|--------|-------|
| MedQA | "A 23-year-old pregnant woman at 22 weeks gestation presents with burning upon urination. She states it started 1 day ago and has been worsening despite drinking more water and taking cranberry extract. She otherwise feels well and is followed by a doctor for her pregnancy. Her temperature is 97.7F (36.5C), blood pressure is 122/77 mmHg, pulse is 80/min, respirations are 19/min, and oxygen saturation is 98% on room air. Physical exam is notable for an absence of costovertebral angle tenderness and a gravid uterus. Which of the following is the best treatment for this patient? A. Ampicillin B. Ceftriaxone C. Doxycycline D. Nitrofurantoin" | "The clinical scenario suggests that the pregnant woman is likely experiencing a urinary tract infection (UTI), given her symptoms of burning upon urination. During pregnancy, the risk of UTIs increases due to physiological changes, and it is important to treat them to prevent complications such as pyelonephritis or preterm labor.
When considering antibiotic therapy for a UTI in a pregnant woman, it's crucial to choose an agent that is safe for both the mother and the fetus. Let's evaluate the options:
A. Ampicillin - This is a penicillin antibiotic that is generally considered safe during pregnancy and can be used to treat UTIs. However, it may not be the first choice due to resistance patterns.
B. Ceftriaxone - This is a cephalosporin antibiotic that is also generally safe in pregnancy and effective against a broad range of bacteria. It is typically used for more complicated infections or when other antibiotics are not suitable.
C. Doxycycline - This is a tetracycline antibiotic that is contraindicated during pregnancy because it can affect fetal bone growth and discoloration of teeth.
D. Nitrofurantoin - This antibiotic is commonly used to treat uncomplicated UTIs and is considered safe during most of the pregnancy. However, it is not recommended at the very end of pregnancy (after 38 weeks) due to the risk of hemolytic anemia in the newborn.
Given that the patient is at 22 weeks gestation and has an uncomplicated UTI, the best choice would be an antibiotic that is safe and commonly used for UTIs in pregnancy.
So, the answer is D. Nitrofurantoin." | D | This question would likely be rated around a 7 or 8. It requires a strong understanding of obstetrics and gynecology, as well as knowledge of common urinary tract infections and their treatment. Additionally, the question involves the consideration of pregnancy-specific treatment options and the potential risks to the fetus. |

| Dataset | Question | GPT4-Answer | Answer | Score |
|---------|----------|-------------|--------|-------|
| MedMCQA | "Chronic urethral obstruction due to benign prismatic hyperplasia can lead to the following change in kidney parenchyma
A. Hyperplasia B. Hyperophy C. Atrophy D. Dyplasia" | "Chronic urethral obstruction, often caused by benign prostatic hyperplasia (BPH), can lead to a series of complications in the urinary tract, especially affecting the kidneys. The obstruction hinders the flow of urine, leading to urinary retention and increased pressure back up into the kidneys. This increased pressure can impair kidney function and eventually lead to changes in the kidney tissue.
The correct term here is ""atrophy."" When there is chronic obstruction, such as from BPH, the kidneys can suffer from prolonged pressure, which leads to the deterioration of kidney tissue over time. This process, where the kidney tissue becomes thinner and weaker, is known as atrophy. Unlike hyperplasia, which is an increase in the number of cells, atrophy refers to the loss or shrinkage of tissue. Hyperophy is not a recognized medical term related to kidney changes. Dysplasia, on the other hand, refers to the presence of abnormal cells within a tissue or organ but does not directly result from obstructive processes like BPH.
So, the answer is C." | C | "Score: 7
Justification: This question requires a deep understanding of medical terminology, anatomy, and the pathophysiology of urinary tract obstructions. It also involves knowledge of the specific changes that occur in kidney parenchyma due to chronic urethral obstruction." |
| MedQA-Evol | "In a 23-year-old pregnant female at 22 weeks gestation with a confirmed urinary tract infection experiencing dysuria, which condition could potentially develop as a direct sequela of this infection if left untreated, and is particularly associated with the immunological adaptations during pregnancy?
A. Gestational diabetes B. Preterm labor C. Intrauterine growth restriction (IUGR) D. Hyperemesis gravidarum" | "In pregnancy, a woman's immune system undergoes various adaptations, making her more susceptible to certain infections, including urinary tract infections (UTIs). If a UTI during pregnancy is left untreated, it can ascend to the kidneys, leading to a more severe infection known as pyelonephritis. This condition can further complicate the pregnancy.
A significant risk associated with untreated UTIs in pregnancy is preterm labor. The inflammation and infection can stimulate the production of prostaglandins, which are chemicals that can induce labor. This can result in preterm birth, where the baby is born before 37 weeks of gestation.
Preterm labor is a well-recognized complication of untreated UTIs during pregnancy and is distinguished from the other options because: - Gestational diabetes is related to hormonal changes affecting insulin, not directly linked to urinary tract infections. - Intrauterine growth restriction (IUGR) refers to poor growth of a fetus while in the mother's womb during pregnancy and is not typically a direct consequence of UTIs. - Hyperemesis gravidarum, characterized by severe nausea and vomiting, is not directly caused by UTIs.
So, the answer is B." | B | "Score: 7
Justification: This question requires a deep understanding of both urinary tract infections and the immunological adaptations during pregnancy. It also involves understanding potential complications of untreated infections during pregnancy. The question is complex and requires a high level of medical knowledge." |

| Dataset | Question | GPT4-Answer | Answer | Score |
|---|---|---|---|---|
| TextBookQA | "In a comprehensive regional anatomical analysis of the thorax emphasizing the respiratory system's components, which structure would NOT be expected to be scrutinized, considering the respiratory focus?
A. External intercostal muscles B. Costovertebral joints C. Visceral pleura D. Coronary vessels" | "The thorax, or chest region, contains several structures, including those related to the respiratory system. When focusing on the respiratory components, one would typically examine structures directly involved in the process of breathing and gas exchange.
A. External intercostal muscles - These muscles are significant in the respiratory system because they are involved in the mechanics of breathing, particularly in the elevation of the ribs during inspiration.
B. Costovertebral joints - These joints are where the ribs articulate with the vertebrae. They are relevant to the respiratory system because they allow the movement of the ribs during respiration.
C. Visceral pleura - This is the membrane that directly covers the lungs. It is definitely a part of the respiratory system, playing a role in the lubrication and protection of the lungs.
D. Coronary vessels - These are the blood vessels of the heart (coronary arteries and veins) that provide the myocardium (heart muscle) with oxygen and nutrients. They are not directly involved in the respiratory system but are part of the circulatory system.
Given that the coronary vessels are not components of the respiratory system but rather of the circulatory system, they would not be expected to be scrutinized in a comprehensive regional anatomical analysis of the thorax emphasizing the respiratory system's components.
So, the answer is D." | D | "Score: 7
Justification: This question requires a deep understanding of regional anatomy of the thorax, specifically focusing on the respiratory system. It involves knowledge of anatomical structures, their relationships, and their functions. The question also requires the ability to identify which structure is not directly related to the respiratory system, adding a level of complexity." |

| Dataset | Question | GPT4-Answer | Answer | Score |
|---|---|---|---|---|
| PubMedQA | "Context: Phosphatidylethanolamine N-methyltransferase (PEMT), a liver enriched enzyme, is responsible for approximately one third of hepatic phosphatidylcholine biosynthesis. When fed a high-fat diet (HFD), Pemt(-/-) mice are protected from HF-induced obesity; however, they develop steatohepatitis. The vagus nerve relays signals between liver and brain that regulate peripheral adiposity and pancreas function. Here we explore a possible role of the hepatic branch of the vagus nerve in the development of diet induced obesity and steatohepatitis in Pemt(-/-) mice. Context: 8-week old Pemt(-/-) and Pemt(+/+) mice were subjected to hepatic vagotomy (HV) or capsaicin treatment, which selectively disrupts afferent nerves, and were compared to sham-operated or vehicle-treatment, respectively. After surgery, mice were fed a HFD for 10 weeks. Context: HV abolished the protection against the HFD-induced obesity and glucose intolerance in Pemt(-/-) mice. HV normalized phospholipid content and prevented steatohepatitis in Pemt(-/-) mice. Moreover, HV increased the hepatic anti-inflammatory cytokine interleukin-10, reduced chemokine monocyte chemotactic protein-1 and the ER stress marker C/EBP homologous protein. Furthermore, HV normalized the expression of mitochondrial electron transport chain proteins and of proteins involved in fatty acid synthesis, acetyl-CoA carboxylase and fatty acid synthase in Pemt(-/-) mice. However, disruption of the hepatic afferent vagus nerve by capsaicin failed to reverse either the protection against the HFD-induced obesity or the development of HF-induced steatohepatitis in Pemt(-/-) mice. Does vagus nerve contribute to the development of steatohepatitis and obesity in phosphatidylethanolamine N-methyltransferase deficient mice? A. maybe B. yes C. no" | "The evidence from the context given indicates that the hepatic branch of the vagus nerve does play a role in the development of diet-induced obesity and steatohepatitis in Pemt(-/-) mice. This is supported by the observation that hepatic vagotomy (HV) abolished the protection against HFD-induced obesity and glucose intolerance in Pemt(-/-) mice. HV also normalized phospholipid content and prevented steatohepatitis, indicating a clear connection between the hepatic vagus nerve and the development of these conditions. However, the disruption of the hepatic afferent vagus nerve by capsaicin treatment did not reverse the protection against HFD-induced obesity or the development of steatohepatitis. This suggests that while the hepatic vagus nerve has a role, it may specifically involve the efferent fibers, which are responsible for sending signals from the brain to the liver, rather than the afferent fibers, which carry signals from the liver to the brain. Based on the provided information, the hepatic branch of the vagus nerve contributes to the development of steatohepatitis and obesity in Pemt(-/-) mice, as evidenced by the effects of HV. However, the specific pathways and mechanisms, whether efferent or afferent, require further clarification. So, the answer is B." | B | This question would be rated as a 9. It requires a deep understanding of hepatic physiology, neurology, and molecular biology, as well as familiarity with the specific roles of phosphatidylethanolamine N-methyltransferase and the vagus nerve in the development of obesity and steatohepatitis. Additionally, the question involves complex experimental design and interpretation of results. |

| Dataset | Question | GPT4-Answer | Answer | Score |
|---|---|---|---|---|
| MedQuad | How to diagnose Lymphocytic Choriomeningitis (LCM) ? | "Diagnosing Lymphocytic Choriomeningitis (LCM) typically involves a combination of clinical assessment, laboratory testing, and, in some cases, imaging studies. Here‚Äôs a detailed approach:
1. **Clinical Assessment**: The initial step is a thorough medical history and physical examination. LCM should be suspected in individuals presenting with symptoms like fever, headache, muscle aches, and meningitis or encephalitis signs, especially if there's a history of exposure to rodents or their droppings, as the LCM virus is primarily transmitted through rodent contact.
2. **Laboratory Testing**: - **Serologic Testing**: This is the primary method for diagnosing LCM. Blood tests can detect antibodies against the LCM virus. IgM antibodies suggest recent infection, while IgG antibodies indicate past exposure. - **Polymerase Chain Reaction (PCR)**: This test can detect LCM virus RNA in blood or cerebrospinal fluid (CSF) and is useful in the early stages of the disease. - **CSF Analysis**: Examination of cerebrospinal fluid obtained via lumbar puncture can show a lymphocytic pleocytosis (increased lymphocytes), which is indicative of meningitis. While not specific to LCM, it can support the diagnosis in the appropriate clinical context.
3. **Imaging Studies**: While not diagnostic for LCM specifically, imaging studies like MRI or CT scans of the brain may be performed to rule out other causes of neurological symptoms.
4. **Differential Diagnosis**: It‚Äôs important to consider and rule out other causes of meningitis or encephalitis, such as other viral, bacterial, or fungal infections, and non-infectious causes like autoimmune diseases.
5. **Epidemiological Links**: In cases where LCM is suspected, identifying a potential source of infection (like contact with rodents) can support the diagnosis.
Remember, the diagnosis of LCM should be made based on a combination of clinical findings, laboratory results, imaging studies (when necessary), and epidemiological links. It‚Äôs also important to consult with healthcare professionals for accurate diagnosis and management." | "During the first phase of the disease, the most common laboratory abnormalities are a low white blood cell count (leukopenia) and a low platelet count (thrombocytopenia). Liver enzymes in the serum may also be mildly elevated. After the onset of neurological disease during the second phase, an increase in protein levels, an increase in the number of white blood cells or a decrease in the glucose levels in the cerebrospinal fluid (CSF) is usually found. Laboratory diagnosis is usually made by detecting IgM and IgG antibodies in the CSF and serum. Virus can be detected by PCR or virus isolation in the CSF at during the acute stage of illness." | This question would likely be rated a 7. It requires a deep understanding of medical terminology and diagnostic procedures, as well as knowledge of the specific symptoms and characteristics of LCM. Additionally, it may involve knowledge of laboratory tests and diagnostic imaging techniques. |

| Dataset | Question | GPT4-Answer | Answer | Score |
|---|---|---|---|---|
| MedInstruct-52k | "Rewrite the following passage into simpler language to help pre-med students understand.
Hypertrophic cardiomyopathy is a condition where the myocardium becomes hypertrophied without an obvious cause, resulting in impaired ventricular filling and sometimes outflow obstruction. The most common presenting symptom is dyspnea on exertion" | Hypertrophic cardiomyopathy is when the heart muscle gets abnormally thick for no clear reason. This thickening can make it hard for the heart to fill with blood properly, and sometimes it can also block blood from flowing out of the heart easily. The symptom people usually notice first is getting short of breath when they do physical activities. | "Hypertrophic cardiomyopathy is a condition where the muscle in the heart becomes too thick without a clear reason, leading to problems with the filling of the heart's chambers and sometimes blocking the flow of blood. The main symptom that most people experience is shortness of breath when they do physical activities.
The Answer is: ""Hypertrophic cardiomyopathy means the heart muscle getting larger for no clear reason, causing trouble with filling the heart and sometimes blocking blood flow. The main symptom is feeling out of breath during exercise.""" | "I would rate this question a 7. Justification: This question requires a deep understanding of medical terminology and concepts related to cardiology. It also requires the ability to simplify complex medical information for a specific audience, adding to its difficulty and complexity." |

| Dataset | Question | GPT4-Answer | Answer | Score |
|---------|----------|-------------|--------|-------|
| Medical-Instruction-120k | Hi.My names Ahdieh.I m from a small city in Iran.My father had a heart attack on sunday as doctor said cpr condition.after 40 minuts he came back to life. now he isnot conscious.he is in ICU.his doctor said his conscious rate is 5.what do you think about his condition?If it is needed I can send you his cardiograf and blood test and the drugs they are using for him in ICU.Kindly tell me can we carry him to a better hospital in Tehran(It takes 4 hours to reach there)? | "Hello Ahdieh, I'm deeply sorry to hear about your father's situation. It must be a very stressful and difficult time for you and your family.
Regarding your father's condition, a Glasgow Coma Scale (GCS) score of 5 indicates severe impairment of consciousness. This scale is commonly used to assess the level of consciousness in individuals who have suffered brain injury or are in a coma. The scale ranges from 3 to 15, with lower scores indicating more severe impairment. A score of 5 suggests that your father is in a critical condition and needs intensive medical care.
I am not a medical doctor, but I can provide some general information. The decision to transfer a patient from one hospital to another, especially one who is critically ill, involves weighing the risks and benefits. Transporting a critically ill patient over a long distance can be risky, and the medical team must be prepared to manage any complications that might arise during the transfer. The potential benefits of being treated at a facility with more advanced resources must be carefully considered against these risks.
Here are some factors to consider:
1. **Current Level of Care**: Is the current hospital equipped to provide the necessary level of care for your father's condition? If not, a transfer might be justified.
2. **Stability for Transfer**: Is your father stable enough for the transfer? Patients must often be stable before they can be transported, especially over long distances.
3. **Medical Transport Services**: If a transfer is necessary, it should be done with appropriate medical transport services equipped to handle critical patients, including the presence of a medical team that can provide care en route.
4. **Receiving Hospital**: Does the hospital in Tehran have the capability to provide better or more specialized care for your father's condition? Have you confirmed that they can accept your father and that they are prepared to provide the necessary treatment?
5. **Consultation with Medical Professionals**: The most important step is to consult with the doctors and medical team currently treating your father. They can provide specific advice based on his condition and the resources available at both the current hospital and the potential receiving hospital in Tehran.
Regarding the cardiogram, blood tests, and medication list, I am not a physician and cannot interpret these medical documents or provide medical advice. It's essential that you discuss these details with your father's medical team. They can provide you with the most accurate information and guidance.
Remember to take care of yourself during this challenging time, and don't hesitate to reach out to family, friends, or support groups for emotional support. Your well-being is also important.
I hope your father's condition improves, and I wish you strength and courage as you navigate this difficult situation." | " Hello Addie! Welcome and thank you for asking on Chat Doctor! I passed carefully through your question and would explain that your father clinical situation could be related to prolonged brain hypoxia, due to the cardiac arrest. The time (40 minutes) needed for his resuscitation is long and may have caused irreversible damage to his brain, due to low blood flow (also called hypoxic encephalopathy). For this reason, I recommend consulting with a neurologist and performing a brain MRI and EEG to examine his brain function. Hope you will find this answer helpful! Kind regards, " | This question would be rated a 9 in difficulty and complexity. It requires a deep understanding of medical terminology, knowledge of CPR and its effects, interpretation of medical tests, and the ability to make a critical decision about transferring a patient to another hospital. Additionally, it involves emotional and ethical considerations. |

| Dataset | Question | GPT4-Answer | Answer | Score |
|---------|----------|-------------|--------|-------|
| ChatDoctor | gyno problemsfor the past few months, I have been having issues with my vagina. there always seems to be something wrong with me. its either an infection or a yeast infection from the medication used to treat the previous infection or a herpes outbreak as a result of a yeast infection. most recently, I had a uti. I was treated for that and everything seemed fine, until after I finished the medication. it still hurt when I had sex and still is uncomfortable to pee. I dont know whats going on and this has been going on for months. | "I'm sorry to hear that you've been experiencing these issues. It's important to remember that I'm an AI and not a medical professional, but I can offer some general advice that might help guide you towards getting the appropriate care.
Recurrent infections and discomfort can be caused by a variety of factors, and it's crucial to address these with a healthcare provider. Here are some steps you might consider:
1. **Follow-Up with Your Healthcare Provider**: Since you're still experiencing discomfort after treatment, it's important to return to your healthcare provider for a follow-up. They might need to conduct further tests to understand why you're having recurrent issues.
2. **Get Tested for STIs**: If you haven't already, make sure to get a full screening for sexually transmitted infections (STIs), including herpes, which you mentioned. This will help to rule out or confirm any infections that may require specific treatment.
3. **Consider Other Causes**: Recurrent urinary tract infections (UTIs) and yeast infections can sometimes be caused by factors like sexual activity, personal hygiene products, antibiotics, hormonal changes, or even underlying health conditions.
4. **Review Your Medications**: If you find that you're getting yeast infections after antibiotic treatments, this could be due to the antibiotics disrupting the natural balance of bacteria and yeast in your vagina. Your doctor might suggest probiotics or other treatments to help maintain this balance.
5. **Discuss Sexual Health**: Pain during intercourse (dyspareunia) can be caused by infections, but it can also be due to other conditions like endometriosis, vaginal dryness, or pelvic inflammatory disease. It's important to discuss this with your healthcare provider.
6. **Lifestyle Adjustments**: Consider lifestyle changes that can reduce the risk of infections, such as wearing breathable cotton underwear, avoiding douches and scented hygiene products, urinating before and after sex, and ensuring good hydration.
7. **Specialist Referral**: If your primary care provider is unable to determine the cause or provide effective treatment, you may need to see a specialist such as a gynecologist or a urologist.
8. **Pain Management**: If you're experiencing ongoing pain, your healthcare provider can also help you with pain management strategies.
9. **Immune System Evaluation**: Since recurrent infections can sometimes be a sign of a weakened immune system, it might be worth evaluating your overall health and immune function.
It's important to address these issues sooner rather than later, as untreated infections can lead to more serious complications. Be open with your healthcare provider about all of your symptoms and concerns so they can provide the most effective care.
Remember, this is general advice and should not replace professional medical consultation. It's essential to work closely with a healthcare provider to get to the root of the problem and find a treatment plan that works for you." | Dear Friend. Welcome to Chat Doctor. I am Chat Doctor. I understand your concern. Recurring yeast / final infection occur due to:1. Improper treatment.2. Inadequate treatment3. Wrong diagnosis.4. Not treating the partner. I would advise you to get a Urine examination done for bacteria and fungus. When the reports are available, get treated. Also, treat your partner. This is my personal opinion for you based on available details. If you have any further query please ask me. Stay Healthy-ChatDoctor, MD | This question would be rated as a 9. It requires a deep understanding of gynecological issues, including infections, medication interactions, and the potential for underlying conditions such as herpes. It also involves analyzing the patient's symptoms and medical history to make an accurate diagnosis. |

| Dataset | Question | GPT4-Answer | Answer | Score |
|---------|----------|-------------|--------|-------|
| WikiInstruct | Evaluate the survival curve graph illustrating the differential survival rates of organ transplants when augmented with regulatory T cell therapy compared to a control group lacking this immunomodulatory intervention, including an analysis of the variance in outcomes. Delve into the prognostic significance of these findings for future graft viability, considering underlying immune response mechanisms. | Evaluating a survival curve that illustrates the differential survival rates of organ transplants augmented with regulatory T cell therapy compared to a control group without this intervention requires considering several key aspects: 1. **Survival Curve Analysis**: The survival curve, often plotted as a Kaplan-Meier curve, would show the proportion of grafts surviving over time for both groups. A higher curve for the group receiving regulatory T cell therapy would indicate better survival rates. It's important to look at the separation between the two curves; a significant gap suggests a meaningful impact of the therapy. 2. **Statistical Significance**: To analyze variance in outcomes, the log-rank test or Cox proportional hazards model might be used to determine if the differences in survival rates are statistically significant. This analysis would help confirm whether the observed differences are likely due to the therapy rather than chance. 3. **Prognostic Significance**: If the survival curve demonstrates significantly better outcomes with regulatory T cell therapy, this has profound implications for future graft viability. It suggests that modulating the immune response via regulatory T cells can effectively prolong graft survival, potentially reducing the need for long-term immunosuppression, which has its own side effects. 4. **Immune Response Mechanisms**: The underlying mechanism involves regulatory T cells' role in maintaining immune tolerance. By augmenting with regulatory T cell therapy, the immune system's attack on the transplanted organ can be minimized. This therapeutic approach targets the immune system's fundamental response to foreign antigens, aiming to establish a more tolerant environment for the graft. 5. **Future Implications**: The success of regulatory T cell therapy in improving graft survival rates could revolutionize transplant medicine by offering a more targeted approach to prevent graft rejection. This could lead to protocols that improve the quality of life for transplant recipients, reduce the incidence of graft-versus-host disease, and possibly decrease the reliance on general immunosuppressants, which have broad and often detrimental effects on the patient's overall immune function. In summary, a survival curve showing improved outcomes with regulatory T cell therapy would highlight the therapy's potential as a game-changer in transplant medicine. It underlines the importance of immune modulation in transplant success and paves the way for research into similar strategies that could further improve graft survival and patient outcomes. | | I would rate this question a 9. The question requires a deep understanding of organ transplantation, immunology, and statistical analysis. It involves a high level of technical terminology and requires a comprehensive understanding of immune response mechanisms and their impact on graft viability. |

| Dataset | Question | GPT4-Answer | Answer | Score |
| --- | --- | --- | --- | --- |

# 7 Croissant Document

We have uploaded our dataset on Huggingface, which automatically generates the following cross-section. For details, please refer to https://huggingface.co/api/datasets/TsinghuaC3I/UltraMedical/croissant.

```json
{
  "@context": {
    "@language": "en",
    "@vocab": "https://schema.org/",
    "citeAs": "cr:citeAs",
    "column": "cr:column",
    "conformsTo": "dct:conformsTo",
    "cr": "http://mlcommons.org/croissant/",
    "data": {
      "@id": "cr:data",
      "@type": "@json"
    },
    "dataBiases": "cr:dataBiases",
    "dataCollection": "cr:dataCollection",
    "dataType": {
      "@id": "cr:dataType",
      "@type": "@vocab"
    },
    "dct": "http://purl.org/dc/terms/",
    "extract": "cr:extract",
    "field": "cr:field",
    "fileProperty": "cr:fileProperty",
    "fileObject": "cr:fileObject",
    "fileSet": "cr:fileSet",
    "format": "cr:format",
    "includes": "cr:includes",
    "isLiveDataset": "cr:isLiveDataset",
    "jsonPath": "cr:jsonPath",
    "key": "cr:key",
    "md5": "cr:md5",
    "parentField": "cr:parentField",
    "path": "cr:path",
    "personalSensitiveInformation": "cr:personalSensitiveInformation",
    "recordSet": "cr:recordSet",
    "references": "cr:references",
    "regex": "cr:regex",
    "repeated": "cr:repeated",
    "replace": "cr:replace",
    "sc": "https://schema.org/",
    "separator": "cr:separator",
    "source": "cr:source",
    "subField": "cr:subField",
    "transform": "cr:transform"
  },
  "@type": "sc:Dataset",
  "distribution": [
    {
      "@type": "cr:FileObject",
      "@id": "repo",
      "name": "repo",
      "description": "The Hugging Face git repository.",
      "contentUrl": "https://huggingface.co/datasets/TsinghuaC3I/UltraMedical/tree/refs%2Fconvert%2Fparquet",
      "encodingFormat": "git+https",
      "sha256": "https://github.com/mlcommons/croissant/issues/80"
    },
    {
      "@type": "cr:FileSet",
      "@id": "parquet-files-for-config-default",
```

```
59        "name": "parquet-files-for-config-default",
60        "description": "The underlying Parquet files as converted by
     Hugging Face (see: https://huggingface.co/docs/datasets-server/
     parquet).",
61        "containedIn": {
62          "@id": "repo"
63        },
64        "encodingFormat": "application/x-parquet",
65        "includes": "default/*/*.parquet"
66      }
67    ],
68    "recordSet": [
69      {
70        "@type": "cr:RecordSet",
71        "@id": "default",
72        "name": "default",
73        "description": "TsinghuaC3I/UltraMedical - 'default' subset\n\
     nAdditional information:\n- 1 skipped column: conversations",
74        "field": [
75          {
76            "@type": "cr:Field",
77            "@id": "default/id",
78            "name": "default/id",
79            "description": "Column 'id' from the Hugging Face parquet
     file.",
80            "dataType": "sc:Text",
81            "source": {
82              "fileSet": {
83                "@id": "parquet-files-for-config-default"
84              },
85              "extract": {
86                "column": "id"
87              }
88            }
89          },
90          {
91            "@type": "cr:Field",
92            "@id": "default/type",
93            "name": "default/type",
94            "description": "Column 'type' from the Hugging Face parquet
     file.",
95            "dataType": "sc:Text",
96            "source": {
97              "fileSet": {
98                "@id": "parquet-files-for-config-default"
99              },
100             "extract": {
101               "column": "type"
102             }
103           }
104         },
105         {
106           "@type": "cr:Field",
107           "@id": "default/answer",
108           "name": "default/answer",
109           "description": "Column 'answer' from the Hugging Face
     parquet file.",
110           "dataType": "sc:Text",
111           "source": {
112             "fileSet": {
113               "@id": "parquet-files-for-config-default"
114             },
115             "extract": {
116               "column": "answer"
117             }
```

```
118                }
119            },
120            {
121              "@type": "cr:Field",
122              "@id": "default/score",
123              "name": "default/score",
124              "description": "Column 'score' from the Hugging Face parquet
       file.",
125              "dataType": "sc:Text",
126              "source": {
127                "fileSet": {
128                  "@id": "parquet-files-for-config-default"
129                },
130                "extract": {
131                  "column": "score"
132                }
133              }
134            }
135          ]
136        }
137      ],
138      "conformsTo": "http://mlcommons.org/croissant/1.0",
139      "name": "UltraMedical",
140      "description": "The UltraMedical Collections\n\t\n\nThe UltraMedical
         Collections is a large-scale, high-quality dataset of biomedical
        instructions, comprising 410,000 synthetic and manually curated
        samples.\nThe construction of UltraMedical adheres to principles
         of diversity and complexity.\nDetailed information can be found in
         our GitHub repository: UltraMedical.\nStatistics of datasets in
        the UltraMedical collections is shown in following table, where...
         See the full description on the dataset page: https://huggingface
        .co/datasets/TsinghuaC3I/UltraMedical.",
141      "alternateName": [
142        "TsinghuaC3I/UltraMedical",
143        "UltraMedical"
144      ],
145      "creator": {
146        "@type": "Organization",
147        "name": "TsinghuaC3I",
148        "url": "https://huggingface.co/TsinghuaC3I"
149      },
150      "keywords": [
151        "100K\u003Cn\u003C1M",
152        "English",
153        "mit",
154        "Croissant",
155        "Region: US"
156      ],
157      "license": "https://choosealicense.com/licenses/mit/",
158      "url": "https://huggingface.co/datasets/TsinghuaC3I/UltraMedical"
159 }
```