# OpenReview forum: "UltraMedical: Building Specialized Generalists in Biomedicine"
_NeurIPS.cc/2024/Datasets_and_Benchmarks_Track — NeurIPS 2024 Track Datasets and Benchmarks Spotlight_

### Official Review · Reviewer_AmEP · 2024-07-14

**Rating:** 7
**Confidence:** 4
**Correctness:** Yes
**Clarity:** Yes

**Review:**

- The UltraMedical collections, featuring both manually curated and synthetic datasets with preference annotations, represent a valuable contribution. These datasets effectively address the scarcity of medical domain-specific data and offer a solid foundation for fine-tuning medical LLMs.

- The authors showcase impressive performance across various medical benchmarks by fine-tuning a suite of specialized medical models based on the Llama-3 series LLMs. This highlights the practical utility and effectiveness of the UltraMedical collections.

- The paper includes comprehensive ablation studies and case studies, providing thorough insights into the method's performance and applications.

**Strengths:**

See above.

**Additional Feedback:**

Overall a useful data source/an open medical LLM. I believe it will be of interest to the medical ML community.

**Documentation:**

Yes

**Limitations:**

Yes

**Opportunities For Improvement:**

The comparison appears to be biased in favor of the proposed method, as it utilizes numerous task-specific labels, whereas many baselines rely on few-shot or even zero demonstrations.

The rationale behind the selection of the backbone model is not discussed. Would using a biomedical pretrained model as the backbone yield better results?

The process of curating the reward benchmark requires further elaboration. For instance, what is the level of agreement among different labelers? How is the quality of the evaluation data ensured?

The paper does not mention the cost associated with creating the benchmark.

**Relation To Prior Work:**

Yes

**Summary And Contributions:**

This paper addresses a pressing need in the development of specialized Large Language Models (LLMs) by introducing the UltraMedical collections, a comprehensive dataset suite designed to enhance fine-tuning in the biomedicine domain. The authors successfully highlight the limitations faced by the open-source community in acquiring specialized data, which hinders advancements in preference learning and other cutting-edge techniques.

---

> ### Author Rebuttal · Authors · 2024-08-15
>
> We sincerely thank you for your positive feedback and valuable suggestions. Below, we address all comments in detail, and we also will revise our paper accordingly.
>
> > The comparison appears to be biased in favor of the proposed method, as it utilizes numerous task-specific labels, whereas many baselines rely on few-shot or even zero demonstrations.
>
> The primary goal of our paper is to demonstrate that our specially constructed datasets can enhance the specialized capabilities of current advanced LLMs, such as Llama, during the post-training phase.
> Existing models like Llama, Mixtral, Qwen, and Deepseek have been finetuned on extensive general domain instructions.
> Our comparison under zero-shot and few-shot settings revealed that the Llama series exhibits the most robust performance.
> Our datasets not only further improve these capabilities, which underscores the effectiveness of our UltraMedical approach, but also show superior performance when benchmarked against other medical LLMs like OpenBioLM and BioMistral.
>
> > The rationale behind the selection of the backbone model is not discussed. Would using a biomedical pretrained model as the backbone yield better results?
>
> Our experiments indicate that the general capabilities of the base model are paramount.
> For instance, "Llama 3 + Medical finetuning" outperforms "Llama 2 + Medical pre-training and finetuning".
> This is likely because the 15 trillion tokens pre-training data of Llama 3 includes almost all available medical corpora.
> Therefore, we opted for the most powerful base models available, aiming to enhance specialized abilities during the post-training period.
>
> > The process of curating the reward benchmark requires further elaboration. For instance, what is the level of agreement among different labelers? How is the quality of the evaluation data ensured?
>
> We have provided initial details about the human annotation process in the supplementary material (Section 4.2), which will be expanded in the revised paper.
> To ensure the reliability of the medical reward benchmark, we assembled a team of three experts, each with at least three years of research experience in biomedicine.
> They utilized a customized WebUI and academic search engines to validate question-answer pairs.
> For the reward benchmark, out of 1,000 test samples, only about 780 were retained where at least two annotators agreed on the same label.
> Samples with disagreements or both incorrect answers were removed.
>
> > The paper does not mention the cost associated with creating the benchmark.
>
> The costs associated with creating the dataset and benchmark primarily include GPT-4-Turbo API calls for instruction synthesis and response generation, as well as preference annotation, totaling approximately $20,000.
> This breakdown will be detailed further in the revised version of the paper.
>
> We appreciate your questions and are eager to further clarify any points or discuss the paper in more detail.

---

> > ### Comment · Reviewer_AmEP · 2024-08-19
> >
> > Thanks for your response. I don't have other questions and have updated the score.

---

### Official Review · Reviewer_QMZQ · 2024-07-23
**Review of UltraMedical: Building Specialized Generalists in Biomedicine**

**Rating:** 8
**Confidence:** 4
**Correctness:** The paper does not have any clear err…
**Clarity:** The paper is well written.

**Review:**

This work presents well-structured contribution to the field of biomedical AI, presenting a comprehensive dataset that result in competitive performance with proprietary models. The methodology is rigorous, combining manual and synthetic data to create a diverse and complex dataset that enhances model training. The development of a medical reward model and iterative preference learning further strengthen the paper's contributions. However, the reliance on GPT-4 for annotations may introduce bias. Despite these minor drawbacks, the paper's significance and clarity make it a valuable resource for advancing open-source biomedical models, supporting the broader research community.

**Strengths:**

The paper demonstrates significant strengths through its comprehensive dataset construction, rigorous fine-tuning methodology, and the development of a medical reward model, all of which contribute to the creation of highly competitive open-source biomedical models. By combining manual and synthetic data, the UltraMedical dataset enhances the diversity and complexity of training instructions, leading to models that perform on par with proprietary counterparts.

**Additional Feedback:**

N/A

**Documentation:**

Documentation is sufficient

**Limitations:**

The authors acknowledge limitations related to the use of GPT-4 annotations and computational constraints in iterative preference learning. They suggest future work to mitigate these issues, such as leveraging open-source models for annotation and exploring more efficient preference learning strategies.

**Opportunities For Improvement:**

1. While the paper is generally clear, including more visual aids or diagrams to illustrate complex processes like iterative preference learning could enhance understanding.
2. In terms or originality, exploring additional innovative techniques for preference learning and reward modeling could further distinguish this work.

**Relation To Prior Work:**

Relation to Prior work is well discussed.

**Summary And Contributions:**

The paper presents the UltraMedical collections, a comprehensive dataset combining manual and synthetic biomedical instructions, used to fine-tune a suite of specialized medical models based on the Llama-3 series. The authors demonstrate the potential of these models through various medical benchmarks, indicating that open-source models can achieve performance comparable to proprietary models like GPT-4 and MedPaLM. The paper's key contributions include the creation of a high-quality dataset, fine-tuning of models to achieve competitive results, development of a medical reward model, and public release of the datasets and models to foster progress in biomedical AI.

---

> ### Author Rebuttal · Authors · 2024-08-15
>
> Thank you for your constructive feedback and effort in reviewing our work. Below, we provide detailed responses to your comments.
>
> > While the paper is generally clear, including more visual aids or diagrams to illustrate complex processes like iterative preference learning could enhance understanding.
>
> Your suggestion is indeed valuable. We initially created a detailed illustration to depict the evolution of the policy model and the annotation of synthetic data.
> Due to page limitations, this figure was condensed and only a small portion included on the first page.
> We plan to expand and update this figure in the revised version of the paper to better illustrate these complex processes.
>
> > In terms or originality, exploring additional innovative techniques for preference learning and reward modeling could further distinguish this work.
>
> The primary aim of this paper is to provide datasets and benchmarks in the specialized biomedical domain, aligning with the goals of the Datasets and Benchmarks track.
> While we acknowledge the importance of developing new algorithms in preference learning for specialized domains, our initial focus is to establish robust datasets and benchmarks. As the proverb goes, "A clever cook cannot prepare a meal without rice."
> Once these foundational resources are established, future work will have the opportunity to explore innovative techniques.
>
> > The authors acknowledge limitations related to the use of GPT-4 annotations and computational constraints in iterative preference learning. They suggest future work to mitigate these issues, such as leveraging open-source models for annotation and exploring more efficient preference learning strategies.
>
> This was our consideration during the submission period.
> With the recent release of advanced open-source models like Llama 3.1 405B [1] and Nemotron-4 340B [2], we now have access to GPT-4 level LLMs in-house.
> The availability of these models opens up new possibilities for our work, allowing us to mitigate biases associated with GPT-4 and address issues related to scope of use and ownership.
> We are excited about the potential to optimize our processes using these advanced models in the future.
>
> [1] Dubey, Abhimanyu, et al. "The llama 3 herd of models." arXiv:2407.21783 (2024).
>
> [2] Adler, Bo, et al. "Nemotron-4 340B Technical Report." arXiv:2406.11704 (2024).

---

> > ### Comment · Reviewer_QMZQ · 2024-08-21
> > **Comment by Reviewer QMZQ**
> >
> > Thanks to the authors for their response. My questions have been addressed. I hope the author add the figure in revised version. I have updated the score.

---

### Official Review · Reviewer_n4GX · 2024-07-23
**Very well-prepared work with large scale dataset and pre-trained models**

**Rating:** 9
**Confidence:** 4
**Correctness:** Yes
**Clarity:** Yes

**Review:**

Very comprehensive and solid work on both dataset and the new fine-tuned model. Especially the preference annotation which plays an important role in the further fine-tuning.

**Strengths:**

1. A new medical dataset to further train the LLM.
2. A fine-tuned LLaMA3 on the new medical dataset.
3. A wide range of LLM models are tested as benchmark including some large scale model.

**Additional Feedback:**

N/A

**Documentation:**

Yes

**Opportunities For Improvement:**

1. Use a separate subsection to discuss the differences between this work and previous works. (Because there are many other related works..)
2. Extend this work to multi-modality in the future.

**Relation To Prior Work:**

Novel, comprehensive datasets with preference annotated. The experiments of tasks and models also covers a wider range.

**Summary And Contributions:**

The paper introduces the UltraMedical collections, which include well curated manual and synthetic biomedical datasets with preference annotations for fine-tuning advanced medical models, leading to outstanding performance on various medical benchmarks and aiding in the development of robust reward models for improved preference learning.

Contributions:

1. The paper introduces the UltraMedical collections, a set of high-quality manual and synthetic datasets specifically designed for the biomedicine domain, equipped with preference annotations to support the advancement of Large Language Models (LLMs) in specialized medical applications.
2. Utilizing these datasets, the authors fine-tuned specialized medical models based on the Llama-3 series, achieving exceptional performance across various medical benchmarks, and developed advanced reward models to enhance online preference learning within the biomedical LLM community.

---

> ### Author Rebuttal · Authors · 2024-08-15
>
> We sincerely thank you for your positive feedback and valuable suggestions. Below, we address all comments in detail, and we also will revise our paper accordingly.
>
> > Use a separate subsection to discuss the differences between this work and previous works. (Because there are many other related works..)
>
> Due to the page limitations of the main text, we initially discussed related technologies in general and specifically for the biomedicine domain in Appendix A.
> We will relocate this discussion to the main body of the paper in the revised version and include updates on some prior works as recommended by the reviewers.
> Additionally, several concurrent works have emerged in the one or two months following our submission, which we will also discuss and compare in the updated paper.
>
> > Extend this work to multi-modality in the future.
>
> We value your suggestion regarding extending our work to multi-modality.
> There are some existing works on instruction tuning for multi-modal biomedicine during the SFT stage, but fewer focus on (iterative) preference learning.
> We plan to explore this direction in our future work and aim to extend our pipeline to multi-modal domains from a theoretical perspective.

---

### Official Review · Reviewer_ipWt · 2024-07-25
**A high-quality manual and synthetic datasets in the biomedicine domain and a multi-stage training framework to build Specialized Generalists in Biomedicine**

**Rating:** 8
**Confidence:** 3
**Clarity:** The paper is well written

**Review:**

The quality of the work is high, as evidenced by the comprehensive and methodical approach to dataset creation and model fine-tuning. The authors provide detailed descriptions of their processes, including the use of supervised fine-tuning, reinforcement learning, and preference optimization.
The clarity of the paper is generally good. The authors effectively communicate the motivation behind their work, the methods employed, and the results achieved.
The authors make a significant contribution to the open-source community by creating high-quality datasets and applying advanced techniques like preference learning. This method, while common, is particularly appreciated for the authors' full disclosure of their open-source data model and code.

**Strengths:**

The research paper meticulously outlines the creation of the UltraMedical dataset, which is a blend of manually crafted and synthetic biomedical instructions. This comprehensive approach ensures that the dataset is both rich in content and diverse in scope. To guarantee the quality and accuracy of the dataset , a rigorous annotation process is employed, involving a collaboration between AI systems and human experts, who together ensure that the biomedical knowledge is correctly captured and categorized.

Furthermore, the paper aims at enhancing the capabilities of open-source language models within the biomedical domain. Supervised fine-tuning, preference learning, and reward modeling are highlighted as effective methods for improving the models' reasoning skills and their ability to perform complex biomedical tasks.

In a thorough evaluation, the UltraMedical models are tested against various medical benchmarks, which serves to demonstrate not only their individual capabilities but also the overall effectiveness of the dataset in driving performance improvements. The authors contribute to the field's transparency and reproducibility by providing detailed accounts of the dataset construction, model training procedures, and evaluation processes.

**Additional Feedback:**

N/A

**Correctness:**

The authors have adequately discussed the limitations and potential negative societal impacts of their work.

**Documentation:**

The authors of this work have provided clear information about data collection and organization, ensuring ethical and responsible use of the data. They have also made the code and data  accessible by providing a GitHub URL and Huggingface URL for reviewers. Additionally, the paper includes a thorough description of the experimental setup used for evaluating benchmarks.

**Ethics:**

I do not see any ethical concerns with the submission

**Limitations:**

Yes, they discussed the possible potential negative societal impact of their work.

**Opportunities For Improvement:**

The paper would benefit from additional information regarding the human annotation process. Specifically, the number of annotators and their expertise in the biomedical field are not clearly stated. Including these details is essential for understanding the context and reliability of the results.

**Relation To Prior Work:**

Apaper worth to be mentioned: Qilin-Med https://arxiv.org/pdf/2310.09089  a multi-stage training method com
bining domain-specific Continued Pre-training (CPT), SFT, and Direct Preference Optimization (DPO). This paper should be cited.

**Summary And Contributions:**

The paper introduces **UltraMedical**, a large-scale dataset tailored for the biomedical domain, containing 410K high-quality instructions derived from both synthetic and manual inputs, along with 100K preference annotations. This dataset is utilized to fine-tune Llama-3 models, combining medical and general domain inputs to create specialized generalists. The study demonstrates the superior performance of these models across various benchmarks, underscoring the necessity of such specialized datasets.

### Contributions
1. **UltraMedical Dataset Construction**:
    - Developed a comprehensive dataset of 410K medical instructions, integrating both synthetic and manual prompts.
    - Annotated 100K instructions with preferences from advanced medical and general models to enhance fine-tuning, reward modeling, and preference learning.

2. **Model Fine-Tuning**:
    - Fine-tuned the Llama-3 series using a multi-step optimization strategy on the UltraMedical dataset.
    - Achieved competitive results in open-source medical benchmarks, narrowing the gap between open-source and proprietary models.

3. **Reward Model Training**:
    - Created a reward model based on UltraMedical preference annotations and previous feedback datasets.
    - Implemented a self-generated response strategy for continuous optimization, resulting in highly effective models.

4. **Benchmark Performance**:
    - Demonstrated that the 8B model significantly outperforms larger models like MedPaLM 1 and GPT-3.5.
    - Achieved the highest score among open-source LLMs on the MedQA-USMLE benchmark with the 70B model, comparable to proprietary models like MedPaLM 2 and GPT-4.

---

> ### Author Rebuttal · Authors · 2024-08-15
>
> Thank you for your positive feedback and diligent efforts in reviewing our paper. We appreciate your comments and would like to share our responses.
>
> > The paper would benefit from additional information regarding the human annotation process. Specifically, the number of annotators and their expertise in the biomedical field are not clearly stated. Including these details is essential for understanding the context and reliability of the results.
>
> Thank you for your suggestion. We have provided some details about the human annotation process in the supplementary material (Section 4.2) and will expand on this in the revised paper.
> To ensure the reliability of the medical rewardbench, we assembled a team of three experts from our cooperative institutions, each holding degrees and possessing at least three years of research experience in biomedicine.
> These experts utilized a customized WebUI and academic search engines to validate the question-answer pairs.
>
> > A paper worth to be mentioned: Qilin-Med https://arxiv.org/pdf/2310.09089 a multi-stage training method com bining domain-specific Continued Pre-training (CPT), SFT, and Direct Preference Optimization (DPO). This paper should be cited.
>
> Thank you for highlighting this key reference. The primary distinction between our UltraMedical and Qilin-Med (referred to as ChiMed) lies in the linguistic domain focus—ours is on English, while the mentioned work focuses on Chinese.
> We will cite this paper and discuss the differences in our revised manuscript.

---

### Decision · Program_Chairs · 2024-09-26

**Decision:**

Accept (Spotlight)

**Comment:**

The paper makes strong contribution to the medical AI community by constructing/sharing comprehensive datasets (manually/synthetically constructed), training and showcasing strong performance of fine-tuned open-source models, and extensive ablations studies.